# Impact of 3D Cloud Structures on the Atmospheric Trace Gas Products from UV-VIS Sounders - Part II: impact on $NO_2$ retrieval and mitigation strategies

Huan Yu[1], Claudia Emde[2], Arve Kylling[3], Ben Veihelmann[4], Bernhard Mayer[2], Kerstin Stebel[3], and Michel Van Roozendael[1]

[1]Royal Belgian Institute for Space Aeronomy (BIRA-IASB), Brussels, Belgium
[2]Ludwig-Maximilians-University (LMU), Meteorological Institute, Munich, Germany
[3]Norwegian Institute for Air Research (NILU), Kjeller, Norway
[4]ESA-ESTEC, Noordwijk, the Netherlands

**Correspondence:** Huan Yu (huan.yu@aeronomie.be)

**Abstract.** Operational retrievals of tropospheric trace gases from space-borne spectrometers are based on one-dimensional radiative transfer models. To minimize cloud effects, trace gas retrievals generally implement a simple cloud model based on radiometric cloud fraction estimates and photon path length corrections. The latter relies on measurements of the oxygen collision pair ($O_2$-$O_2$) absorption at 477 nm or on the oxygen A-band around 760 nm to determine an effective cloud height.

In reality however, the impact of clouds is much more complex, involving unresolved sub-pixel clouds, scattering of clouds in neighboring pixels and cloud shadow effects, such that unresolved three-dimensional effects due to clouds may introduce significant biases in trace gas retrievals. Although clouds have significant effects on trace gas retrievals, the current cloud correction schemes are based on a simple cloud model, and the retrieved cloud parameters must be interpreted as effective values. Consequently, it is difficult to assess the accuracy of the cloud correction only based on analysis of the accuracy of

the cloud retrievals, and this study focuses solely on the impact of the 3D cloud structures on the trace gas retrievals. In order to quantify this impact, we study $NO_2$ as a trace gas example, and apply standard retrieval methods including approximate cloud corrections to synthetic data generated by the state-of-the-art three-dimensional Monte Carlo radiative transfer model MYSTIC. A sensitivity study is performed for simulations including a box-cloud, and the dependency on various parameters is investigated. The most significant bias is found for cloud shadow effects under polluted conditions. Biases depend strongly

on cloud shadow fraction, $NO_2$ profile, cloud optical thickness, solar zenith angle, and surface albedo. Several approaches to correct $NO_2$ retrievals under cloud shadow conditions are explored. We find that air mass factors calculated using fitted surface albedo or corrected using the $O_2$-$O_2$ slant column density can partly mitigate cloud shadow effects. However, these approaches are limited to cloud-free pixels affected by surrounding clouds. A parameterization approach is presented based on relationships derived from the sensitivity study. This allows identifying measurements for which the standard $NO_2$ retrieval produces a significant bias, and therefore provides a way to improve the current data flagging approach.

produces a significant bias, and therefore provides a way to improve the current data flagging approach.

# 1 Introduction

Satellite observations in the UV and visible spectral ranges are widely used to monitor trace gases in the troposphere. Current sensors (GOME-2, OMI, and the newest TROPOMI) as well as future atmospheric Sentinels from the European Copernicus program observe several key tropospheric species, such as $NO_2$ (Boersma et al., 2018; van Geffen et al., 2020; Liu et al., 2020),
HCHO (De Smedt et al., 2018; ?), $SO_2$ (Theys et al., 2015, 2017), and CHOCHO (Lerot et al., 2010). These observations provide important information on fossil fuel combustion emissions, biomass burning, biogenic production, and volcanic emissions and they are highly relevant for the study of air quality and climate change.

In the UV and visible spectral ranges, the main retrieval algorithm is the Differential Optical Absorption Spectroscopy (DOAS) technique (Platt and Stutz, 2008), which consists of two steps: First, the slant column density (SCD) is retrieved by
means of spectral fitting methods involving the direct solar spectra, the Earth reflected solar spectra and laboratory absorption cross-sections of trace gases. The SCD corresponds to the integrated trace gas concentration along the light path taken by photons at the wavelength corresponding to the fitting window, as they travel from the Sun, through the atmosphere and back to the satellite sensor. To convert the SCD into a vertical column density (VCD) one uses air mass factors (AMF) calculated with a radiative transfer model (RTM). The AMF is defined as the ratio of the atmospheric SCD and VCD. In clean regions, the
error of the trace gas retrieval is dominated by the DOAS spectral fitting, while the uncertainty of the AMF becomes important for polluted regions. In general, AMFs depend on a number of factors, including surface albedo, cloud and aerosol properties, as well as the a priori profile shape of the measured trace gas.

Clouds have a strong influence on the retrieval of the trace gases. Since the UV-visible sensors mentioned above have a relatively coarse spatial resolution, ranging from $3.5\times5.5$ km$^2$ to $40\times80$ km$^2$, only a small percentage of the observed
pixels (10-20%) are cloud free (Krijger et al., 2007), and most pixels are either fully or partly cloudy. Thus trace gas retrieval algorithms rely on cloud property information provided for each ground pixel. Such information is important, since clouds have a significant impact on the photon path. The effect of clouds on the trace gas retrieval has been studied by several authors (e.g. Boersma et al., 2004; Lorente et al., 2017; Liu et al., 2020). In these studies, the cloud treatment is based on the independent pixel approximation (IPA). A simple cloud correction scheme is generally used, which treats clouds as Lambertian surfaces or
scattering layers, and relies on the concepts of cloud fraction, cloud top albedo and cloud top pressure (Acarreta et al., 2004; Wang et al., 2008; Loyola et al., 2018).

In order to correct for the presence of clouds in the trace gas retrievals, several approaches to the cloud retrieval are described in the literature. They are based on the determination of the mean photon path in the visible and near-infrared (NIR) bands from analysis of a spectral feature of a well-mixed species. For example, the $O_2$-$O_2$ cloud retrieval uses the 477 nm absorption
band of the oxygen collision pair (Acarreta et al., 2004; Sneep et al., 2008; Stammes et al., 2008; Veefkind et al., 2016). The Fast Retrieval Scheme for Clouds from the $O_2$-A band (FRESCO) algorithm uses reflectance measurements around the $O_2$-A band (Koelemeijer et al., 2001; Wang et al., 2008). The Optical Cloud Recognition Algorithm and the Retrieval Of Cloud Information using Neural Networks (OCRA/ROCINN) retrieve the cloud fraction from analysis of the broadband colour of the measured spectra, and the cloud top albedo and cloud top height from the $O_2$-A band (Loyola et al., 2007, 2018). The $O_2$-$O_2$

cloud product has been applied to the $NO_2$ retrieval from OMI (Boersma et al., 2007, 2011; Bucsela et al., 2006, 2013). The operational products developed at DLR for GOME-2 and TROPOMI use the OCRA/ROCINN cloud algorithm (Valks et al., 2011; Theys et al., 2017; De Smedt et al., 2018; Liu et al., 2019), while the FRESCO cloud algorithm developed at KNMI has been used for trace gas retrievals from GOME, SCIAMACHY, GOME-2 and TROPOMI (Boersma et al., 2004, 2018; van Geffen et al., 2021).

The retrieval of trace gases from space sensors is performed using one-dimensional (1D) radiative transfer models. However, cloudy scenes are influenced by 3D structures and the impact of 3D features like spatial heterogeneities and structured cloud boundaries increases when the spatial resolution of the instruments approaches the dimensions of cloud features. Therefore, measurements by space sensors like TROPOMI and the future Sentinel-4 and Sentinel-5, which are designed to resolve horizontal features equal or better than $7{\times}7$ km$^2$, will be strongly influenced by 3D clouds. Nikolaeva et al. (2005) summarizes the effects introduced by 3D clouds but not captured by 1D radiative transfer:

(1) Shadowing effect: decreased reflectance within the cloud geometric shadow.

(2) Channelling effect: channelling of photons from the cloud to the cloud-free (shadow) side, which leads to the increased reflectance near the cloud.

(3) Leaking effect: photons leaking at the cloud edge, which decreases reflectance near the border of the cloud (inside the cloud).

(4) Brightening effect: increased reflectance at cloud edges that are directly illuminated by the Sun.

Several studies have demonstrated the presence of 3D cloud effects in satellite observations. For example, Várnai and Marshak (2009) examined the clear sky reflectance enhancements near clouds based on MODIS observations. The enhancements are apparent at distances less than 15 km to nearest clouds, and are stronger at shorter wavelengths and near optically thicker clouds. Várnai et al. (2013) examined the retrieval of aerosols near low-level maritime clouds using co-located MODIS and CALIOP observations. These results indicate that the 3D radiative processes contribute to near-cloud reflectance enhancements, especially within 1 km from clouds. Massie et al. (2017, 2021) provided observational evidence of 3D cloud effects in OCO-2 $CO_2$ retrievals based on analysis of OCO-2 column-averaged $CO_2$ data combining with MODIS radiance and cloud fields. The impact of 1D assumptions has not been well explored in trace gas retrievals from satellite UV-visible sensors, however, the recent studies by Schwaerzel et al. (2020, 2021) demonstrated the importance of 3D effects on airborne and ground-based measurements.

This paper is one of a series of three papers discussing the impact of 3D cloud structures on the atmospheric trace gas products from satellite UV-visible sounders. One by Emde et al. (2022) describes the generation of MYSTIC synthetic data used for validation of 1D trace gas retrieval algorithms, and another one by ? identifies and quantifies possible 3D cloud related retrieval bias based on both synthetic and observational data. The present paper focuses on impact of 3D effects on the classic tropospheric trace gas retrievals, including identification and investigation of the significant retrieval biases due to the 3D clouds, and exploration of mitigation strategies for these cases.

The 3D effects affect the cloud retrievals first and then the trace gas retrievals, and in this study, the main focus is on the influence of 3D clouds on the trace gas retrievals. In order to investigate this impact, we study $NO_2$, a key tropospheric trace gas measured by atmospheric Sentinels. In Section 2, we first describe our standard DOAS retrieval algorithm, which includes a simplified cloud correction approach. Based on these tools, Section 3 presents a sensitivity study of the $NO_2$ retrieval for synthetic 2D box-clouds. The dependency on various parameters is studied and the scenarios giving the most significant biases are identified. We then investigate which parameters can be extracted from synthetic 3D cloud simulations and correlated to retrieval biases. Finally, in Section 4, several mitigation strategies are explored and applied to both synthetic and observed data.

## 2 Methodologies

### 2.1 Computation of the tropospheric AMF

The standard DOAS method assumes that the retrieved slant column can be converted into a vertical column using an AMF $M$, which accounts for the average light path of the light through the atmosphere. For an optically thin absorber (typically the optical thickness $\tau_{NO_2} \sim 0.0025 \ll 1$ for $5 \times 10^{15}$ molec./cm$^2$ of $NO_2$ column at 460 nm), the trace gas has a negligible effect on the radiation field, and the AMF can be written as a linear sum of the altitude-dependent AMF of each layer, weighted by the $NO_2$ partial vertical column density (Palmer et al., 2001):

$$M = \frac{\sum_l m_l \cdot x_l}{\sum_l x_l} \tag{1}$$

where $x_l$ is the $NO_2$ partial column density for layer $l$. The altitude-dependent AMF $m_l$ is calculated in the same way as the total air mass factor, but for an optically thin amount of trace gas in layer $l$ only. The tropospheric AMF is computed as the integral of layer $l$ from the ground up to the tropopause. Notice that in previous studies (e.g. Lorente et al., 2017) the altitude-dependent AMF was referred to as box-AMF. However, in order to distinguish the box-AMF from 3D simulation, we will use the term layer-AMF for 1D simulation.

The AMF is computed using radiative transfer calculations that require information on measurement conditions (such as observation geometry and wavelength) and atmospheric characteristics (e.g., vertical distribution of the species, surface albedo and clouds). Hence, an appropriate selection of the a priori assumptions used is essential to obtain the correct values of the AMF and thus reduce the uncertainties of the $NO_2$ retrieval. Selecting an AMF too large will result in an underestimation of the VCD. Likewise, the determined $NO_2$ VCD will be too large if the value of the AMF used for the conversion of the SCD is too small.

### 2.2 Cloud correction

To correct for cloud effects on trace gas retrievals, a simple approach is usually used. The AMF for a partly cloudy scene is determined using the IPA (Boersma et al., 2004), which assumes that the AMF can be written as a linear combination of a

cloudy and a clear-sky AMF:

$$M = (1 - cf_w) \cdot M_{clr}(A_s, P_s) + cf_w \cdot M_{cld}(A_c, P_c) \tag{2}$$

Where $A_c$ and $A_c$ represent surface albedo and cloud top albedo, $P_s$ and $P_c$ are surface pressure and cloud top pressure. $M_{clr}$ is the AMF for a cloud-free scene, and $M_{cld}$ is the AMF for a fully cloudy scene. The intensity weighted cloud fraction (CF$_w$) $cf_w$ is defined as:

$$cf_w = \frac{cf_r \cdot R_{cld}(A_c, P_c)}{cf_r \cdot R_{cld}(A_c, P_c) + (1 - cf_r) \cdot R_{clr}(A_s, P_s)} \tag{3}$$

where $cf_r$ is the radiometric cloud fraction (CF$_r$). $R_{clr}$ and $R_{cld}$ are the averaged top-of-atmosphere (TOA) reflectances over the fitting interval for a clear and a cloudy scene, respectively.

In this study, the cloud properties (radiometric cloud fraction $cf_r$ and effective cloud top pressure $P_c$) are derived by cloud retrieval algorithms based on the collision-induced absorption by oxygen (O$_2$-O$_2$) around 477 nm and the absorption by O$_2$-A band (FRESCO). Both cloud algorithms assume that cloud is a Lambertian reflecting surface with a fixed high albedo of 0.8, and the treatment of clouds is achieved through the IPA, which is consistent with the assumption for the calculation of the AMF. Notice that, all cloud effects, including the 3D effect, are treated based on such simplified cloud correction schemes, however, these approaches may not capture all cloud effects, which leads to uncertainty in the NO$_2$ retrieval.

Aerosols are not included in this study. However, the presence of aerosol may lead to different impacts on the 3D effects, depending on aerosol properties, such as single scattering albedo, optical depth, and vertical distribution. For example, scattering aerosols in the cloud shadow will increase the AMF and compensate the shadowing effect, whereas strong absorbing aerosols may decrease the AMF and increase the 3D effect. The resulting effect may be rather complex, and further investigation would be needed for an accurate evaluation of such effects. In addition, it should be noted that, in practice, aerosols are implicitly treated as clouds in actual retrievals since the effects of aerosols are expected to be similar to those of clouds (Boersma et al., 2004, 2011).

### 2.2.1  O$_2$-O$_2$ cloud retrieval

The O$_2$-O$_2$ cloud retrieval algorithm (Acarreta et al., 2004; Veefkind et al., 2016) is based on the O$_2$-O$_2$ absorption band at 477 nm, and the retrieval consists of two main steps: first, a DOAS fit is performed in the spectral region between 425 nm and 495 nm to derive the O$_2$-O$_2$ slant column amount $S_{O_2\text{-}O_2}$. In the second step the $S_{O_2\text{-}O_2}$ and the TOA reflectance $R$ in the middle of the fit window (460nm) are converted into cloud fraction $cf_r$ and cloud pressure $P_c$ using the following equations:

$$R = (1 - cf_r) \cdot R_{clr}(A_s, P_s) + cf_r \cdot R_{cld}(0.8, P_c) \tag{4}$$

$$S_{O_2\text{-}O_2} = (1 - cf_w) \cdot S_{O_2\text{-}O_2}^{clr}(A_s, P_s) + cf_w \cdot S_{O_2\text{-}O_2}^{cld}(0.8, P_c) \tag{5}$$

where $cf_w$ is computed based on Eq. 3. $R_{clr}$ and $R_{cld}$ are the TOA reflectances for a clear and a cloudy scene, respectively, and $S_{O_2\text{-}O_2}^{clr}$ and $S_{O_2\text{-}O_2}^{cld}$ are the corresponding O$_2$-O$_2$ SCDs. In practice, these parameters are pre-calculated with a radiative

transfer model in the form of a look-up table (LUT), which is a function of solar zenith angle, viewing zenith angle, relative azimuth angle, surface albedo and surface pressure.

For a given geometry, we first compute $S^{cld}_{O_2\text{-}O_2}(0.8, P_c)$ for all possible cloud pressure values (from 0 to $P_c$, referred to as $P^{'}_c$) and save it as $S^{'}_{O_2\text{-}O_2}$. Then, we set $P_c$ = surface pressure $P_s$ for the starting estimation, and take the following steps:

(1) The radiometric cloud fraction is obtained by: $cf_r = \frac{R - R_{clr}(A_s, P_s)}{R_{cld}(0.8, P_c) - R_{clr}(A_s, P_s)}$

(2) The intensity weighted cloud fraction $cf_w$ is calculated using Eq. 3.

(3) $O_2$-$O_2$ SCDs for cloudy scene are derived by: $S^{cld}_{O_2\text{-}O_2} = \frac{S_{O_2\text{-}O_2} - (1 - cf_w) \cdot S^{clr}_{O_2\text{-}O_2}(A_s, P_s)}{cf_w}$

(4) $P_c$ is retrieved from $S^{cld}_{O_2\text{-}O_2}$ using a linear interpolation based on relationship between $P^{'}_c$ and $S^{'}_{O_2\text{-}O_2}$.

In the visible band, $R_{cld}(0.8, P_c) \approx 0.8$ (Stammes et al., 2008), and depends only weakly on cloud pressure. Therefore, the radiometric cloud fraction retrieval does not rely on the cloud pressure retrieval, and the above inversion procedure provides sufficient retrieval accuracy. A further iteration is made by repeating the above steps with the retrieved $P_c$ to get a more accurate result. In order to avoid extrapolation, the inversion process is terminated when $R > R_{cld}(0.8, P_c)$ or $S_{O_2\text{-}O_2} > S^{cld}_{O_2\text{-}O_2}(0.8, P_s)$. In addition, $cf_r = 0$ when $R < R_{clr}(A_s, P_s)$ or $S_{O_2\text{-}O_2} < S^{clr}_{O_2\text{-}O_2}(A_s, P_s)$.

## 2.2.2 FRESCO cloud retrieval

The FRESCO algorithm is based on the absorption in the $O_2$ A-band around 760nm (Koelemeijer et al., 2001; Wang et al., 2008). Cloud pressure and cloud fraction are derived from reflectance measurements at three 1-nm wide windows: namely 758–759 nm, 760–761 nm and 765–766 nm. These represent respectively the continuum window, and stronger and weaker $O_2$ absorption bands. The radiative transfer model used is based on the IPA: the TOA reflectances are computed as the weighted sum of the reflectances of the cloud-free and the cloudy parts of the pixel:

$$R = (1 - cf_r) \cdot A_s \cdot T_{clr} + cf_r \cdot A_c \cdot T_{cld} + (1 - cf_r) \cdot R_{clr} + cf_r \cdot R_{cld} \tag{6}$$

Where $T_{clr}$ and $T_{cld}$ are the direct transmissions along the photon path, and $R_{clr}$ and $R_{cld}$ are the single Rayleigh scattering reflectance including $O_2$ absorption between the surface/cloud and TOA. The transmissions $T_{clr}$ and $T_{cld}$ depend on solar zenith angle, viewing zenith angle, wavelength and pressure level, and include $O_2$ absorption and Rayleigh extinction. The transmissions is calculated using a line-by-line method with the line parameters from the HITRAN2012 molecular spectroscopic database (Rothman et al., 2013), and then convolved using the instrumental spectral response function at the measurement wavelength grid. The retrieval method is based on minimizing the difference between the measured and simulated spectra in the three windows using a Levenberg-Marquardt non-linear least squares method.

## 2.3 Synthetic data

In order to investigate the effect of 3D cloud features on the $NO_2$ retrieval from space sensors, the 3D Monte Carlo model MYSTIC (Mayer, 2009; Emde et al., 2011), which is operated as one of several radiative transfer solvers in the libRadtran

**Table 1.** Settings for the 1D simulation.

| Parameter [units] | Abbreviation | Values |
|---|---|---|
| Solar zenith angle [°] | SZA | 20, 30, 40, 50, 60, 70, 80 |
| Viewing zenith angle [°] | VZA | 0, 30, 60 |
| Relative azimuth angle [°] | RAA | 0, 90, 180 |
| Surface albedo [ ] | ALB | 0, 0.02, 0.05, 0.1, 0.2, 0.3, 0.5, 0.8 |
| Cloud optical thickness [ ] | COT | 0, 1, 2, 5, 10, 20 |
| Cloud bottom height [km] | CBH | 1, 3, 10 |

package (Mayer and Kylling, 2005; Emde et al., 2016), is used to generate synthetic observations. The dataset includes simulated spectra in two spectral ranges (in the visible band from 400-500 nm and in the $O_2$-A band from 755-775 nm). In addition, it includes layer AMFs calculated at 460 nm (for further details see: Emde et al., 2022).

The simulations are calculated based on the US-standard atmosphere (Anderson et al., 1986). The Rayleigh scattering cross section is computed using the parameterization by Bodhaine et al. (1999). For the visible band, the absorptions from $NO_2$, $O_3$ and $O_4$ are taken into account, and the spectra recorded at sampling intervals of 0.2 nm. For the $O_2$-A band, line-by-line simulations are performed with a spectral resolution of 0.005 nm. The absorption coefficients are calculated using the ARTS model (Eriksson et al., 2011) with line parameters from the HITRAN2012 dataset. The simulated spectra are convolved with

a Gaussian response function of Full width at half maximum (FWHM) equal to 0.5 nm, sampled at intervals of 0.2 nm, and finally averaged over three spectral bands: 758-759 nm, 760-761 nm, and 765-766 nm, which are used by the FRESCO cloud retrieval.

There are three groups of datasets generated by MYSTIC:

The first one includes a 1D simulation with a 1-km thick cloud layer for a variety of solar-satellite geometries, surface

albedos, and cloud properties as listed in Table 1. This dataset is used to investigate the uncertainty of the $NO_2$ retrieval due to the simplified cloud correction approaches. In addition, clear sky spectra (COT=0) are calculated for all geometries and surface albedos in order to check the agreement between MYSTIC and VLIDORT RTMs (see Section 2.4).

The second dataset includes a simple box-cloud with a variety of geometrical and optical thickness. The simulation is performed for a nadir viewing sensor with a $1\times1$ km$^2$ field-of-view (FOV) along a line starting at a distance of 15 km away

from the cloud edge in the clear region and ending at a distance of 10 km from the cloud edge in the cloudy scene. This dataset is used to investigate the sensitivity of the $NO_2$ retrieval bias for clear pixels located nearby clouds, and to identify the parameters correlated to 3D effects. Furthermore, possible mitigation approaches are investigated using this dataset.

Finally the third dataset includes realistic three-dimensional clouds and typical geometries representative for Low Earth Orbit (LEO) and Geostationary Earth Orbit (GEO) satellite observations. The cloud field is taken from the Large Eddy Simulation

(LES) based on the ICOsahedral Non–hydrostatic atmosphere model (ICON) (Dipankar et al., 2015; Zängl et al., 2015) for a region including Germany and parts of other surrounding countries. The simulations include all cloud types typical for central Europe. This dataset is used to validate the mitigation approaches described in Section 4 below.

## 2.4 Radiative transfer model settings

Two radiative transfer models are used for the impact assessment of 3D clouds on trace gas retrievals. The synthetic datasets with 3D cloud fields are generated using MYSTIC, whereas the layer-AMFs and modelled reflectances at TOA used for $NO_2$ retrieval and cloud correction are simulated with the linearized vector code VLIDORT (Spurr et al., 2001; Spurr and Christi, 2014, 2019) version 2.7. VLIDORT applies the discrete ordinates method to generate simulated radiances at TOA and analytic derivatives (jacobians) with respect to atmospheric and surface parameters (i.e. weighting functions). The layer-AMFs $m_l$ are derived from altitude-dependent weighting functions determined by VLIDORT:

$$m_l = \frac{\partial \ln I}{\partial \tau_l} = (\tau_l \cdot \frac{\partial I}{\partial \tau_l})/(I \cdot \tau_l) \tag{7}$$

where $I$ is the simulated TOA radiance, $\tau_l$ is the absorption optical thickness of $NO_2$ at layer $l$, and the term $\tau_l \cdot \frac{\partial I}{\partial \tau_l}$ is the altitude-dependent weighting function for $NO_2$.

We first need to ensure consistency between VLIDORT and MYSTIC, therefore an intercomparison exercise was performed for a 1D plane-parallel clear sky atmosphere. The simulations from both models use the same atmosphere including Rayleigh scattering as well as absorption by gases. The comparison of reflectances and layer-AMFs was made for a variety of combinations of solar and viewing geometries and surface albedos as shown in Figure 1.

Figure 1a compares the reflectance at 460 nm and in three wavelength bands (758-759 nm, 760-761 nm, and 765-766 nm) around the $O_2$-A band for all geometries and surface albedos. The overall differences are 0.0007, 0.0002, 0.0001 and 0.0001 for the above four wavelengths. Corresponding relative differences are generally less than 0.5%, except for low surface albedo (0.05) at 760-761 nm where the difference reaches 1%. Figure 1b shows the comparison of the simulated layer-AMFs at 460 nm for all geometries for surface albedos of 0.05 and 0.8. The averaged difference is within 0.5%/0.2% with a standard deviation of 1.8%/0.7% for surface albedo=0.05/0.8. The bias slightly decreases with altitude. The total AMF is calculated from the layer-AMFs by weighting it with two atmospheric absorber profiles: a tropospheric $NO_2$ profile corresponding to a highly polluted case, and a $O_2$-$O_2$ profile from the US-standard atmosphere (Anderson et al., 1986). The tropopause height is set to 15 km in this study. Results are displayed in Figure 1c. The agreement between the models is good with average differences of 0.45% and 0.3% for $NO_2$ and $O_2$-$O_2$.

In the present work, the main focus is on the effect of 3D clouds. Therefore, radiative transfer model settings in the $NO_2$ and cloud retrievals are made as consistent as possible with those used to generate the synthetic data sets. Although some errors are inevitable, such as those related to differences between MYSTIC and VLIDORT, or due to interpolation in the LUTs, these errors are generally small. We are therefore confident that the differences between retrieved $NO_2$ values and truth (as imposed in the synthetic data) mainly come from the simplified cloud correction approach used in the calculation of the AMF and from 3D cloud effects.

In addition, for very low cloud fraction cases ($CF_r$<1%), the cloud top height output is highly unstable, and a small difference between the RTMs will lead to a large uncertainty in the cloud height retrieval. Therefore, it is reasonable to consider the observation with $CF_r$<1% as a clear-sky pixel (i.e., $CF_w$ is set 0 in Eq. 2) in order to avoid unnecessary error propagation

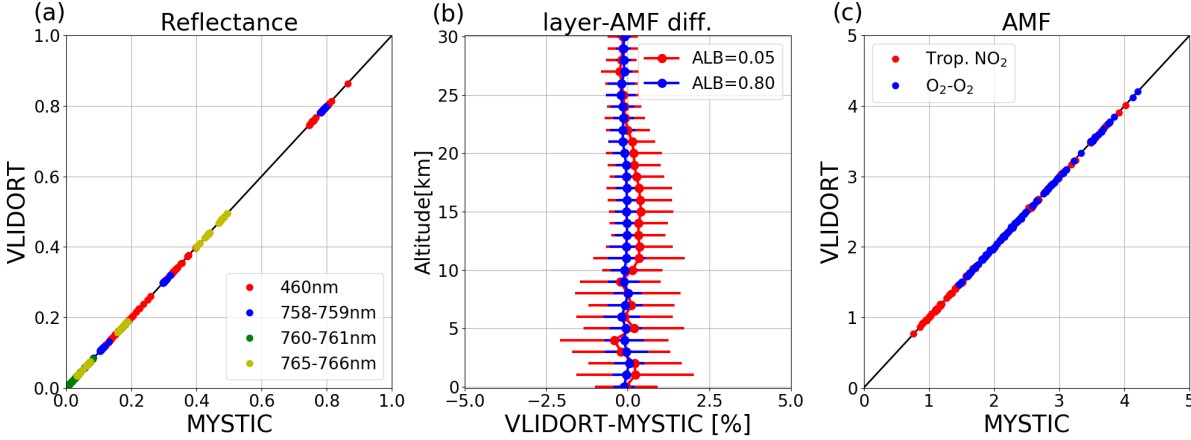

**Figure 1.** Comparison of radiative transfer models (MYSTIC and VLIDORT). (a) TOA reflectance simulated at 460 nm, 758-759 nm, 760-761 nm, 765-766 nm. (b) Relative difference of layer-AMFs. Red (albedo=0.05) and blue (albedo=0.8) circles with error bars (standard error) are calculated for a variety of geometries. Relative difference between a and b is calculated using (a-b)/b·100% herein. (c) comparison of AMF calculated with a highly polluted tropospheric $NO_2$ profile (red) and an $O_2$-$O_2$ profile (blue).

through the retrievals, which can be as high as 10%. Moreover, the cloudy scenes ($CF_w$>50%) are usually excluded in the analysis.

## 2.5  $NO_2$ retrieval for 1D clouds

In this section, we assess the order of magnitude of the uncertainty that is inherent to conventional cloud correction schemes. We use this uncertainty in order to put in perspective the errors due to the simplistic treatment of clouds for scenes with complex 3D clouds. Two conventional cloud correction schemes are considered here, including FRESCO and the $O_2$-$O_2$ cloud correction scheme. The uncertainty inherent to these schemes is assessed for synthetic scenes with known 1D clouds, considering the deviation of air mass factor obtained by these schemes from the synthetic truth (obtained by MYSTIC), and the difference in the air mass factors between the two schemes.

The retrieval algorithm is applied to synthetic data for 1D cloud scenes with the selected SZAs (30°, 60°), VZAs (0°, 30°, 60°), RAAs (0°, 90°, 180°), ALBs (0.05, 0.1, 0.3) and various cloud parameters: 1-km thick cloud with CBH of 1/3/10 km and COT of 1/2/5/10/20. Examples of cloud and $NO_2$ retrievals are shown in Figure A1. The $O_2$-$O_2$ and FRESCO cloud fraction retrievals show very good agreement. However, cloud pressure retrievals show large differences, especially for high cloud cases. It should be noted that the cloud pressure retrievals based on $O_2$-$O_2$ or $O_2$ absorption must be interpreted as effective values. Furthermore, a more accurate cloud retrieval does not always correspond to a better cloud correction in the $NO_2$ retrieval. For instance, the $O_2$-$O_2$ cloud pressures substantially differ from true values for the high cloud cases, whereas FRESCO cloud pressures are usually compared to the middle of the cloud layer. On the other hand, $NO_2$ AMFs using an $O_2$-$O_2$ correction are often closer to the true AMF than those using FRESCO correction. These results also show different

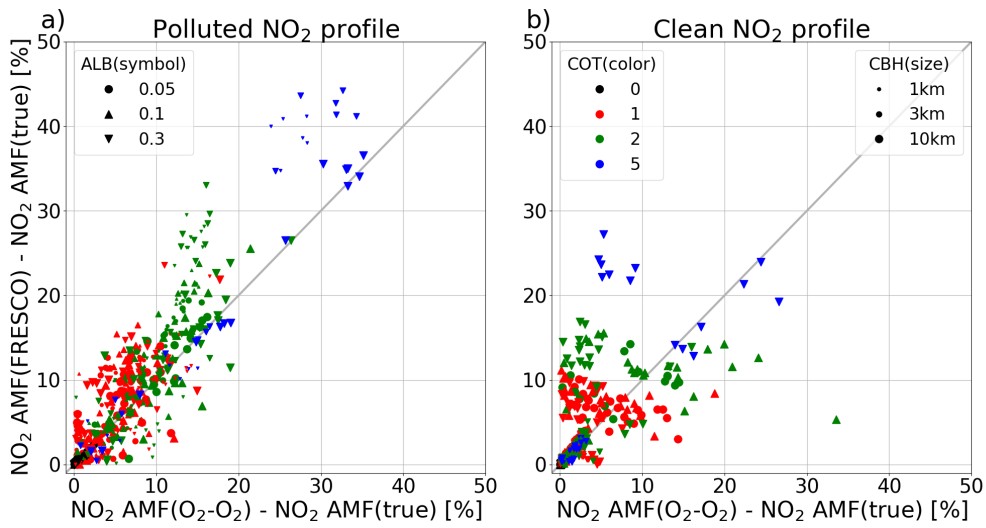

**Figure 2.** Comparison of bias of $NO_2$ AMF retrieval using the cloud correction based on $O_2$-$O_2$ and FRESCO clouds. The retrievals are based on (a) the European polluted and (b) the clean atmospheric $NO_2$ profile, and the retrievals are applied when $CF_w \leq 50\%$. A variety of symbols/colors/marker size represent the cases with the different surface albedo/cloud optical thickness/cloud bottom height.

impact on the retrieval between the polluted and clean cases. It implies that the accuracy of the cloud correction relies not only

255 the accuracy of the cloud retrieval, but also other factors, such as the $NO_2$ profile.

The error of the $NO_2$ retrieval is evaluated by comparing the calculated AMF with the true AMF, which is calculated using layer-AMFs from MYSTIC (see the companion paper by Emde et al., 2022) combined with the $NO_2$ profile. Figure 2 compares the bias of the $NO_2$ AMF retrieval corrected by cloud parameters derived from the FRESCO and $O_2$-$O_2$ algorithms. The retrievals are applied for polluted and clean $NO_2$ profiles, both taken from the CAMELOT study (Levelt et al., 2009).

Retrievals for COT>5 are not shown in the figure, since the corresponding $CF_w$s are larger than 50%, and the cloudy pixels are excluded from the analysis.

The $NO_2$ AMF retrieval using FRESCO and $O_2$-$O_2$ cloud corrections generally shows a good agreement and differences mostly are within 10%, see Figure 2. For the polluted cases (Figure 2a), the bias of the $NO_2$ retrieval is mostly within 20%. Some higher biases occur for pixels having a high surface albedo (0.3). We also observe that retrieval biases obtained using the

265 FRESCO cloud correction are systematically higher than those obtained using the $O_2$-$O_2$ cloud correction. For clean conditions (Figure 2b), the retrieval generally shows a lower bias, except a few cases for high clouds (CBH=10km).

In this study, the calculation of $NO_2$ AMFs assumes perfect knowledge of all parameters, and in particular, the $NO_2$ profile is assumed to be the same inside and outside of the cloud. The error of the $NO_2$ retrieval is mainly from the cloud correction. The bias of the $NO_2$ retrieval using the classic cloud correction schemes is generally lower than 20%. Therefore, this value is

270 used as a reference amplitude to define the significance of 3D effects in the study.

## 3 NO$_2$ retrieval in the vicinity of a box-cloud

### 3.1 Sensitivity study

In reality, the cloud-affected scenes are usually complex, many cloud effects come together that is difficult to distinguish. Moreover, the NO$_2$ retrieval of our interest is (nearly) cloud-free scene. In order to investigate the influence of the different 3D cloud effects on NO$_2$ retrievals, we start with simple box cloud cases, and investigate the NO$_2$ retrievals for the clear pixels around the clouds. Emde et al. (2022) performed MYSTIC radiative transfer simulations with a box-cloud. The simulations are made for an imaginary nadir viewing sensor with a $1 \times 1$ km$^2$ FOV, and two types of cloud base cases are defined to represent a low-altitude liquid cloud (2-3 km) and a high-altitude ice cloud (9-10 km). In addition, the scenarios include a variety of solar zenith angles, surface albedos, cloud optical thickness, cloud geometric thickness (CGT) and cloud bottom heights.

The standard NO$_2$ retrievals based on both O$_2$-O$_2$ and FRESCO cloud algorithms are applied to the synthetic spectra for a polluted case, and the impact of 3D effects is identified on clear sky pixels by comparing AMF values from the retrieval with corresponding true values. Figure 3 shows the bias of the NO$_2$ AMF retrieval due to cloud in-scattering and shadowing. In the in-scattering region (Figure 3a), a negative or positive bias is observed for a few pixels next to the cloud edge. For these pixels, the retrieved CF$_r$ is greater than 0 due to the enhanced reflectance, and the O$_2$-O$_2$ value is slightly larger than that of FRESCO. Cloud pressure retrieval is usually a bit lower than surface pressure, but higher than neighboring cloud pressure, and the FRESCO cloud pressure is relatively higher (not shown). Although there are some differences between the retrievals using O$_2$-O$_2$ and FRESCO cloud corrections, the biases are generally small. In the cloud shadow region, the reflectance is lower than the clear sky reflectance. Accordingly, the retrieved CF$_r$ is 0, and the calculated AMF corresponds to the clear sky AMF. Since the true AMF is generally smaller than the clear sky AMF in the cloud shadow, the retrieved AMF tends to be overestimated (see Figure 3b), and these differences can reach up to 125% depending on the SZA, cloud height, and distance from the cloud edge. Outside of the cloud shadow region, a small retrieval bias remains, especially for the low cloud cases, which is due to an effect of horizontal scattering from the cloud edge (namely, channeling effect). The retrieval biases are generally small for a clean profile as shown in Figure A2 except for the high cloud cases with SZA equal to 80°.

Although cloudy pixels are not our primary focus here, it is interesting to note that retrieval biases for such pixels depend on the distance from the cloud edge, and imply the effect of 3D clouds. Note also that we obtain very good agreement between the retrievals corrected by the two cloud approaches, and only a slightly larger difference (10%) occurs for SZA=80° in the cloud shadow cases.

### 3.2 Identification of conditions leading to the largest biases

In order to study the dependence of the NO$_2$ AMF bias due to the cloud shadowing/in-scattering for the parameters defined in the previous section, the largest absolute retrieval bias over the clear region is selected for each scenario, and is plotted as function of various parameters. The retrieval includes the O$_2$-O$_2$ and FRESCO cloud correction, and the results are shown in Figure 4.

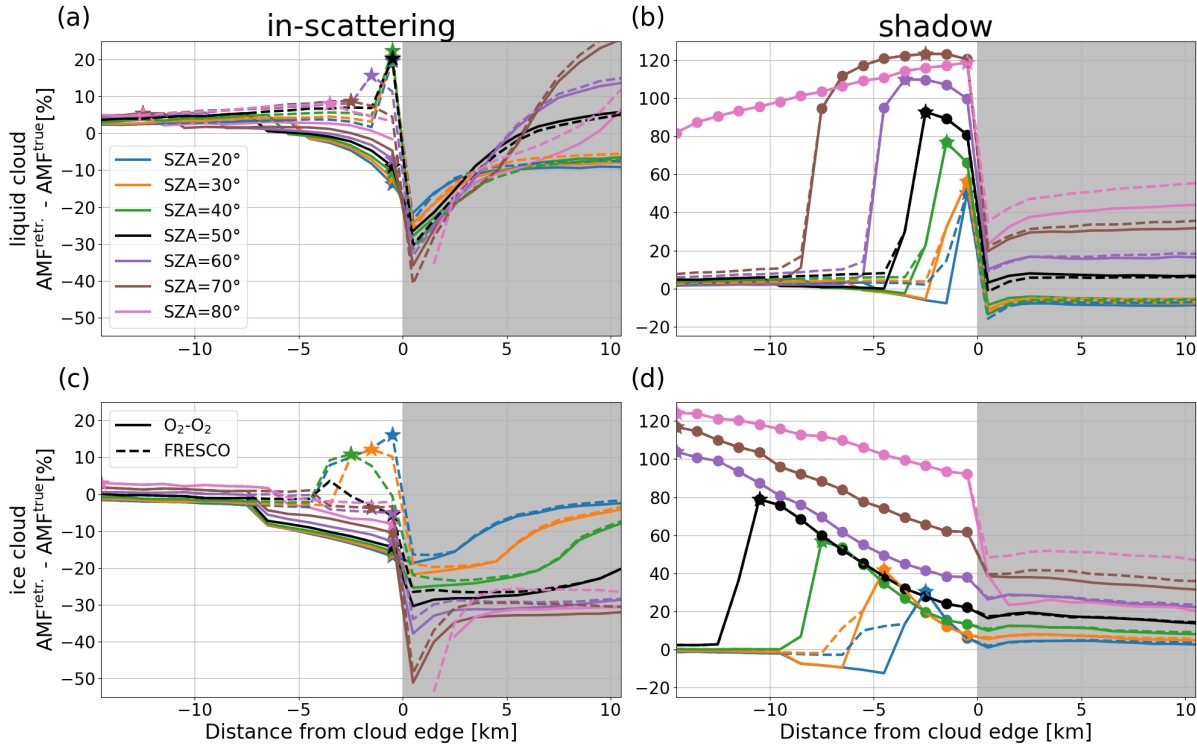

**Figure 3.** NO$_2$ AMF retrieval bias as a function of the distance from the cloud edge for the different SZAs. Negative distances from the cloud edge correspond to the pixels in the clear region (white regions), and positive distances correspond to the pixels in the cloudy region of the domain (gray regions). The top panels are for the low cloud and the bottom for the high cloud. The left panels show cloud in-scattering and the right panels show cloud shadow. Solid and dashed lines correspond to retrievals corrected by O$_2$-O$_2$ and FRESCO cloud algorithms, respectively. Stars correspond to the largest absolute bias over the clear region for each scenario, and dots in the cloud shadow region (b and d) denote the horizontal extent of the cloud shadow.

In the cloud shadow cases, the retrieved CF$_r$ is 0, and therefore the NO$_2$ retrieval does not correct for the presence of clouds. The impact of the cloud shadow strongly depends on the SZA, ALB, and COT. Related biases increase from ~40% for
SZA=20° to more than 100% at high SZA (>60°), and from 10% for COT=0.2 to 120% for COT=20. They decrease from 80-90% for ALB=0.02 to 20% for a higher albedo value (0.3). Increased surface albedos increase the reflection from the ground, which compensates the reduced transmission of sunlight in the cloud shadow and thus reduces relative biases. The dependence of the bias on CGT is relatively small within the range of 50% and 100%, and the impact marginally depends on CBH. In the cloud in-scattering regions, the retrieval biases are much smaller. The retrieval AMFs corrected by O$_2$-O$_2$ and FRESCO cloud
algorithms display biases of up to 25% for all cases. The same analysis was conducted for a clean NO$_2$ profile as shown in Figure A3. In this case, biases are overall small and mostly within 20%. Thus, in the following, we will concentrate on the retrievals in the cloud shadow region for polluted conditions, which give the largest 3D-related biases.

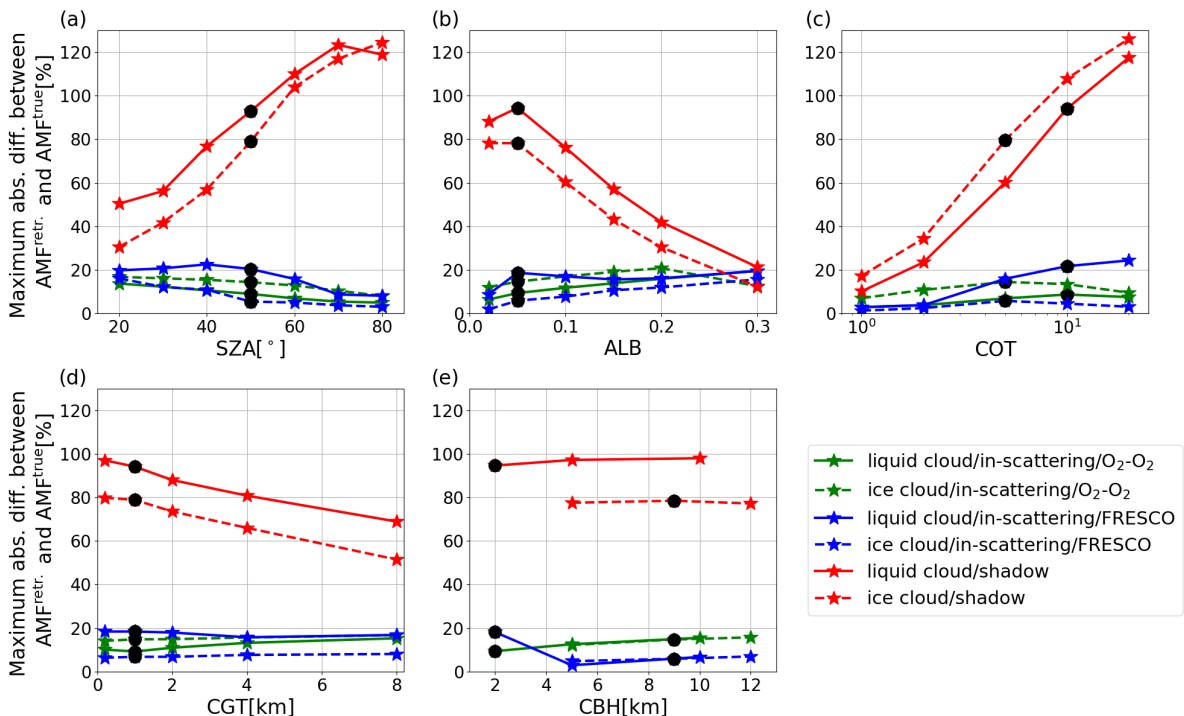

**Figure 4.** Maximum NO$_2$ AMF bias for the polluted NO$_2$ profile in the clear regions as a function of solar zenith angle (a), surface albedo (b), cloud optical thickness (c), cloud geometric thickness (d) and cloud bottom height (e). Solid and dashed lines represent the retrieval for the simulations with liquid and ice water clouds respectively. The green and blue lines depict the AMF biases using O$_2$-O$_2$ and FRESCO cloud corrections over the in-scattering region, and the red lines correspond to the retrieval bias in the cloud shadow. Black dots refer to the base cases (SZA=50°, ALB=0.05, COT=10/5, CGT=1km, CBH=2/9km for liquid/ice cloud), which are defined in Section 3.1 of Emde et al. (2022).

### 3.3 Influence of the NO$_2$ vertical profile

In order to investigate the effect of the NO$_2$ profile on the retrieval, two model profiles with maxima at different heights are used. The box profile has a constant NO$_2$ concentration below the given height, while for the triangle profile, the NO$_2$ concentration decreases linearly with altitude and the value above the given height is 0. Figure 5 shows examples of the box and triangle model profiles with a height of 3 km, as well as the polluted and clean profiles used in the study. The profiles are normalized by the tropospheric columns. They are used to calculate both retrieval and true AMFs, for the cases corresponding to box clouds at different altitudes. The largest retrieval bias of each case is selected as a function of the model profile height and displayed in Figure 5b.

In order to describe the shape of the NO$_2$ profile, we introduce a parameter: the profile height, i.e., the altitude (pressure) below which resides 75% of the integrated tropospheric NO$_2$ profile. For example, the profile height for 3 km box and triangle profiles is 2.25 and 1.5 km, respectively. The bias of the NO$_2$ retrieval for both profile shapes shows a consistent dependency

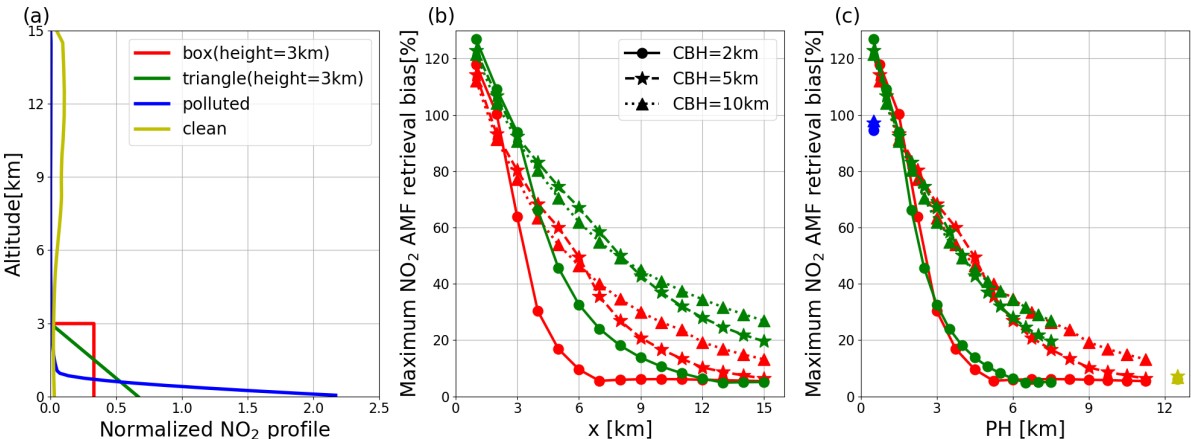

**Figure 5.** Dependence on NO$_2$ profile shape of the NO$_2$ AMF bias in the cloud shadow. (a) selected NO$_2$ profile shapes. (b) largest AMF retrieval biases for the cases with liquid water cloud at different altitudes, as a function of the model height. (c) similar to (b), but as a function of the profile height parameter. See text for further details.

on profile height (Figure 5c). The profile height for the polluted and the clean NO$_2$ profile is 0.5 km and 12.5 km, respectively, and the corresponding NO$_2$ retrieval biases are 95% and 6%. Note that the retrieval bias for the polluted NO$_2$ profile (blue points) is 20-30% lower than for the 1 km box profile, while both profiles share the same profile height. This may link to other factors not considered here, such as the cloud top height. Generally speaking, 3D effects will increase the layer-AMF above the clouds, and decrease it below the clouds (see Figure 6 of Emde et al., 2022). Because of such compensating effects, the presence of NO$_2$ above the cloud will reduce the bias in the AMF calculation for the polluted profile.

## 3.4 Change of spatial resolution

3D cloud effects depend on the spatial resolution of the satellite measurements. The synthetic data with a box-cloud used in this study correspond to a resolution of $1 \times 1$ km$^2$, while the spatial scales of TROPOMI ($3.5 \times 7$ km$^2$ at nadir, $3.5 \times 5.5$ km$^2$ since 6 August 2019), Sentinel-4 (from $9 \times 12$ km$^2$ at a reference point at 45°N, and degrades away from the sub-satellite point) and Sentinel-5 ($7.3 \times 7.5$ km$^2$ at nadir) are larger. In order to investigate 3D effects at the spatial resolution of the Sentinels, we bin synthetic spectra by a factor of 3, 5, 7, 9, 11, 13, 15, to represent the measurements with spatial resolutions of 3-15 km. The new spectra are obtained using moving averages of 3-15 pixels, and the true layer-AMFs are calculated using an intensity-weighted average based on the radiance at 460 nm.

The standard retrieval algorithm using the O$_2$-O$_2$ cloud correction is applied to the binned dataset. Figure 6 shows examples of the NO$_2$ retrieval error based on the binned data for a variety of SZAs and for spatial scales of 3, 7, 11 and 15 km. The pixels can be divided into three categories: (1) the dark gray region on the right side is the cloudy scene, (2) the region on the left side is the clear scene, and (3) the light gray area in the middle part corresponds to partly cloudy partly clear scenes. In the clear region, the number of pixels completely in the cloud shadow (denoted by dots) decreases with the increasing pixel

size. At 3 km resolution, pixels completely in the cloud shadow can be found for SZA$\geq$50°, while such pixels are only found for SZA=80° for a pixel size of 15 km. This is linked to the cloud shadow area, which is determined by the cloud top height and SZA. The retrieval bias significantly decreases when the cloud shadow fraction is less than 1 (pixels on the left side of the dots).

We apply the standard retrieval algorithm to all binned dataset, and extract the same statistics as in Figure 4. Results are shown in Figure 7. In general, the retrieval bias decreases with increasing spatial scales due to spatial averaging. The cloud shadow effect strongly depends on the fraction of the pixel that is in the cloud shadow. When the shadow area is smaller than the size of the satellite footprint, the cloud shadow effect will be significantly reduced. Otherwise, the change is relatively small. The cloud shadow area for the low liquid cloud cases is usually less than 15 km, the AMF retrieval bias significantly decreases with the increasing pixel size. Whereas the dependency of the bias on spatial resolution is relatively weak for the high cloud cases, since their cloud shadow area is usually larger than 15 km. Note that the synthetic data used in this study assumes that the $NO_2$ column is the same in clear and cloudy regions as well as in cloud shadow. Consequently, the $NO_2$ retrieval is based on the same assumption. In reality, however, the $NO_2$ column usually shows significant to large horizontal variability, which leads to uncertainty in the retrieval. The importance of such effects cannot be easily assessed using tools available for this study, and would need to be further investigated.

## 3.5 Cloud shadow fraction

As discussed in the previous section, the retrieval bias significantly decreases when the cloud shadow fraction is less than 1. Therefore, the cloud shadow fraction (CSF) is a key parameter to quantify cloud shadow effects. In order to study the relationship between retrieval bias and cloud shadow fraction, we first extract all the pixels in the clear region from the liquid cloud cases for SZAs of 50° and 70°. Simulations with the different bins are used in the analysis. The cloud shadow fraction is calculated based on the geometric relationship between cloud top height, SZA and distance from the edge of the cloud. Results are shown in Figure 8a. Note that the AMF biases and the cloud shadow fractions are nearly linearly dependent.

In addition, a similar analysis (displayed in Figure 8b) is performed for the partly cloudy region. The colors represent the geometric cloud fraction, and the black points are the averaged retrieval bias in the cloudy and cloud shadow regions. There is an almost linear dependency for most of the pixels. However, some obvious outliers can be found for SZA=70° and CSF=0.55/0.63/0.75. This may be linked to the different contributions of cloudy, shadow and clear sky.

Based on the above discussion, the independent pixel approximation can be used to estimate the retrieval bias. We assume that the bias can be expressed as a linear combination of the bias from the clear sky, shadow, and cloudy parts, and we apply this approach to the data shown in Figure 8b. It should be noted that the retrieval bias is negligible for cloud free pixels since differences between the VLIDORT and MYSTIC models are very small (see Section 2.4). Therefore, the retrieval bias is set to 0 for clear sky scenes. Results are shown in Figure 9. As can be seen, there is a good general agreement between the true bias and the estimated one.

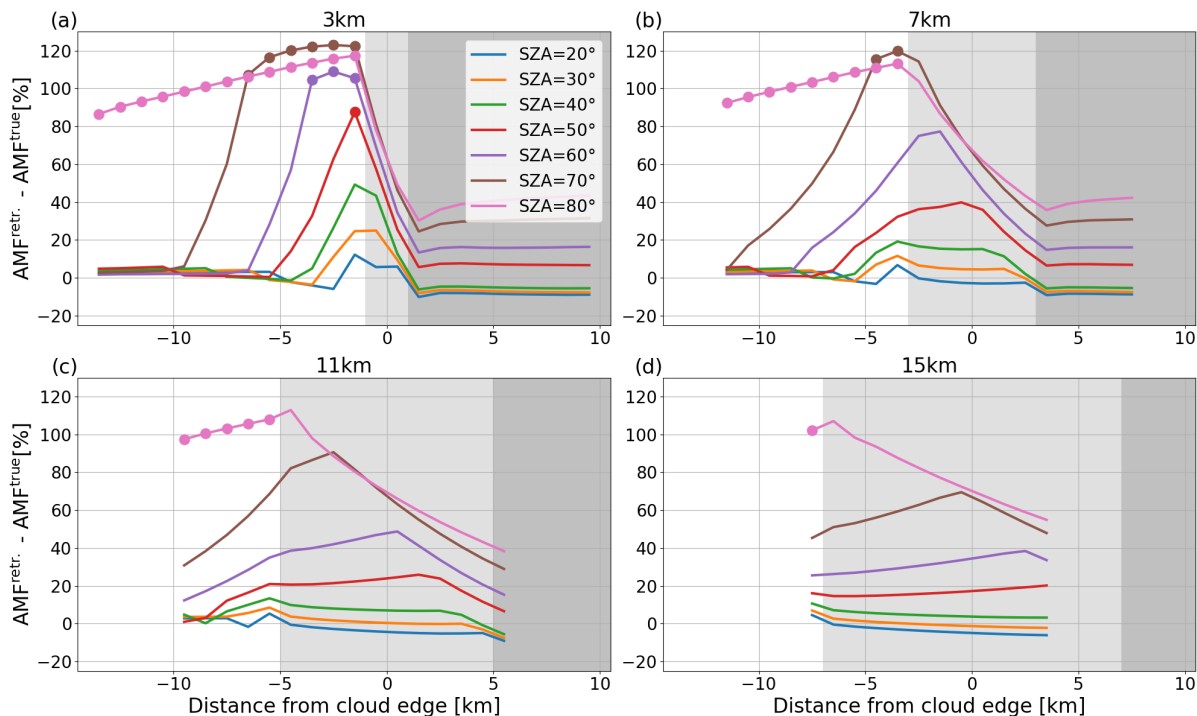

**Figure 6.** Liquid cloud $NO_2$ AMF retrieval bias for box cloud simulations with spatial resolutions of 3(a), 7(b), 11(c), 15(d) km, as a function of the distance from the cloud edge for a variety of SZAs. The dark grey region is fully cloudy, the light gray region partly cloudy, and the white region fully clear. Dots represent conditions where the whole pixel is in the cloud shadow. The AMF uses the $O_2$-$O_2$ cloud correction and is calculated with the polluted $NO_2$ profile.

## 3.6 Dependence on slant cloud optical thickness

We introduce the slant cloud optical thickness (SCOT), which corresponds to the integrated extinction of the cloud from the Sun through the atmosphere to the ground along the line of sight. The SCOT can be used to judge whether a ground pixel is in the cloud shadow. For the box-cloud cases, the SCOT for the pixels in the cloud shadow is calculated as: SCOT = COT / cos(SZA). As we can see in Figure 4, the $NO_2$ bias strongly depends on SZA and COT, which are both linked to the SCOT.

In Figure 10a, the averaged retrieval bias is calculated over the cloud shadow region for each case as a function of SCOT. There is a quasi-linear relation between the bias and the logarithm of the SCOT. The analysis is also made for the synthetic data with the LES clouds. Simulations for nadir observations (VZA=0°), a variety of SZAs (20°, 40° and 60°), and surface albedo of 0.05 are used. The SCOT is calculated from the direct transmittance using MYSTIC based on the synthetic input of 3D fields of the cloud optical thickness from ICON. This approach is the same as for the calculation of the cloud shadow index, which is described in Section 3.3 of **?**. Figure 10b shows the AMF retrieval bias as a function of the SCOT. Again, only

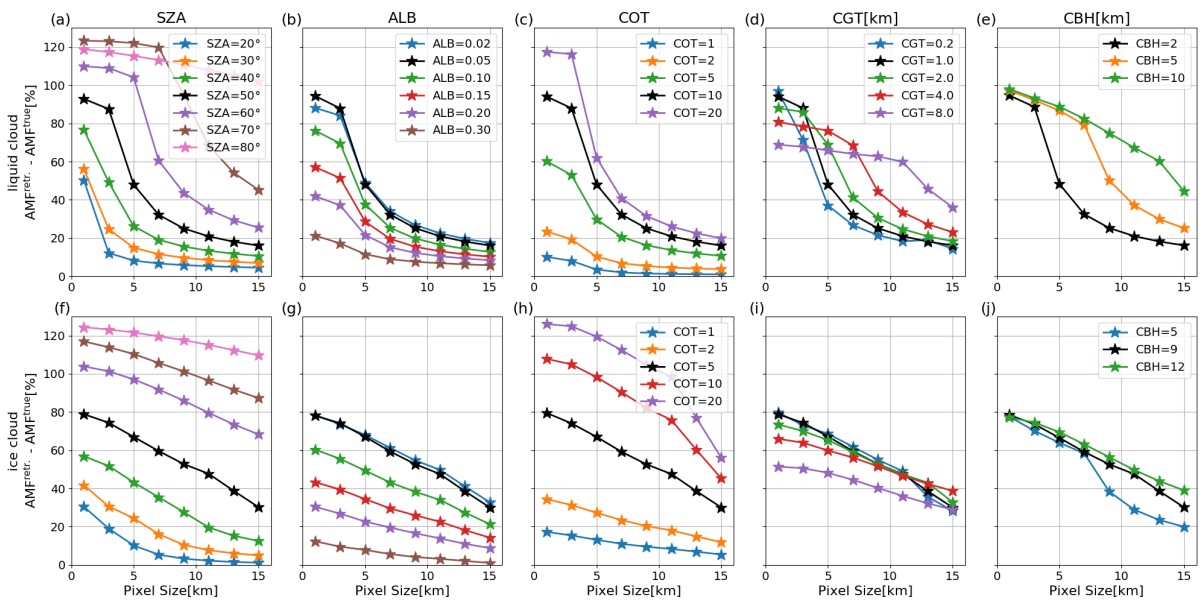

**Figure 7.** Maximum NO$_2$ AMF bias in the cloud shadow as a function of the pixel size for the liquid (2-3 km altitude) and ice clouds (9-10 km altitude) for various values of the SZA, ALB, COT, CGT and CBH.

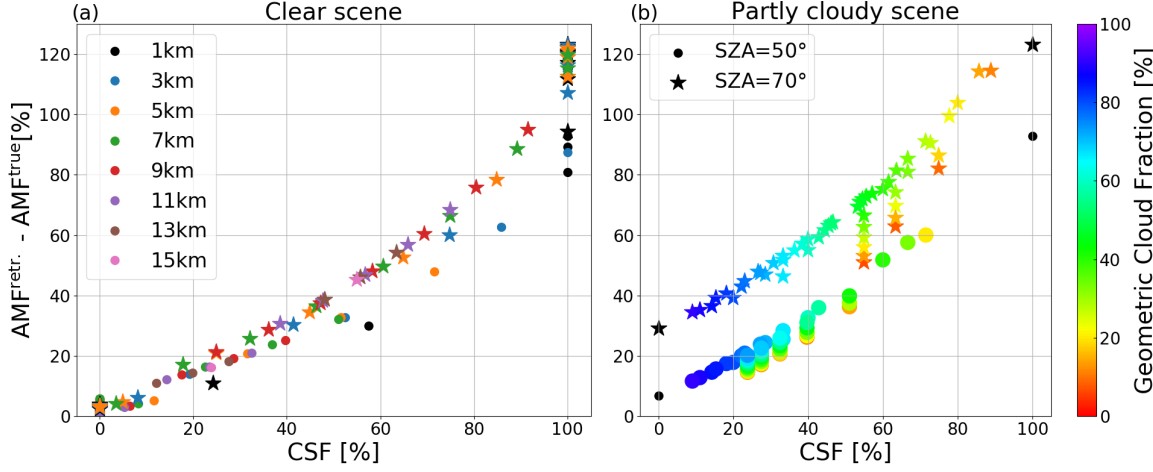

**Figure 8.** NO$_2$ AMF retrieval bias for the liquid cloud cases in the cloud shadow with various spatial resolutions over the clear (a) and the partly cloudy (b) region, depending on cloud shadow fraction. Circles and stars are the cases for SZA=50° and 70°, respectively. See text for further details.

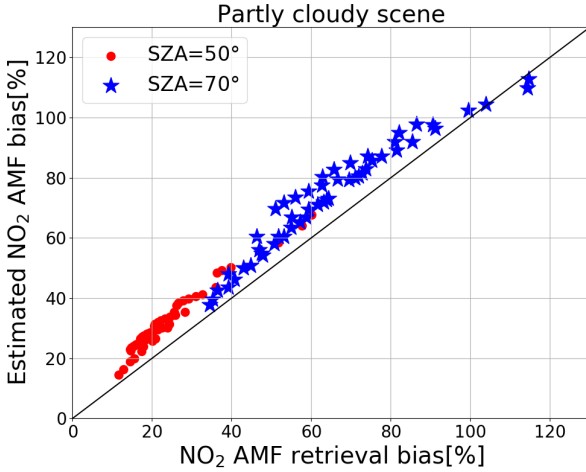

**Figure 9.** Estimated vs. true AMF retrieval bias for partly cloudy scenes. The estimation is based on a linear combination of the AMF retrieval bias over clear, cloud shadow and cloudy scenes. See text for further explanations.

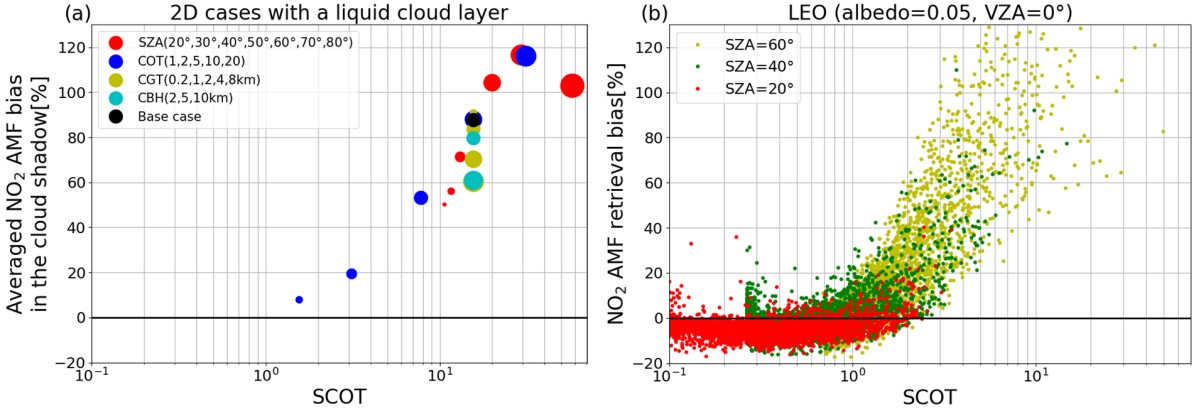

**Figure 10.** $NO_2$ AMF retrieval bias as a function of slant cloud optical thickness. (a) Box-cloud cases with the liquid cloud. The bias is averaged over all the pixels in the cloud shadow, and the various colors and marker sizes represent cases with different solar zenith angles, cloud optical thicknesses, cloud geometric thicknesses and cloud bottom heights. (b) Synthetic data with LES clouds for Low Earth Orbit (LEO) satellite geometries (VZA=0°, SZA=20°, 40° and 60°) and a surface albedo of 0.05. The black line shows the bin average with standard deviations (error bars). Only retrievals with $CF_w$<50% are used.

the nearly cloud-free pixels are used ($CF_w$<50%). The retrieval error is close to 0 when SCOT<1 and significantly increases for SCOT>1.

## 4 Mitigation

### 4.1 Approaches

In this section, various approaches to mitigate biases due to the cloud shadows are explored. These include: (1) calculation of the AMF using an effective isotropic surface albedo that is fitted based on the observed TOA Earth radiance; (2) correction of the $NO_2$ retrieval by using the deviation of the retrieved $O_2$-$O_2$ SCDs and the reference calculations for a clear scene under the same geometry and surface albedo; (3) estimation of the $NO_2$ bias using an empirical formula which parameterizes the bias as a function of driving parameters including the cloud shadow fraction, SCOT, $NO_2$ profile height and cloud top height.

### 4.1.1   $NO_2$ AMF using extended cloud retrievals

For pixels in the cloud shadow, the standard $NO_2$ retrieval will treat the scene as cloud-free when the measured radiance is smaller than the corresponding clear sky radiance. For such pixels, cloud correction is not applied in the AMF calculation. The current cloud algorithms can in fact deal with such situations if the retrieval is extended such that negative cloud fractions are allowed. Figure 11 shows examples of extended $O_2$-$O_2$ (a) and FRESCO (c) cloud retrievals and corresponding $NO_2$ AMFs. In
the cloud shadow regions, both $CF_r$ are negative. The FRESCO $CF_r$ is slightly smaller than the $O_2$-$O_2$ one, while the FRESCO cloud pressure is higher than the $O_2$-$O_2$ one. In addition, the retrieved cloud pressure in the cloud shadow area is higher than the cloud pressure from the neighboring cloud pixels. The bias of the $NO_2$ AMF using the extended $O_2$-$O_2$ cloud is higher than the bias based on the standard approach, whereas this bias is significantly reduced when the AMF calculation uses the extended FRESCO cloud.

In order to further verify these correction approaches, we applied the cloud and $NO_2$ retrievals to the various box cloud scenarios discussed in Section 3, and corresponding results are shown in Figure 11b and Figure 11d. In the cloud shadow regions, the retrieval biases are even higher than the standard retrieval bias when the correction uses the extended $O_2$-$O_2$ retrieval, however they are mostly reduced for the retrievals based on the extended FRESCO retrieval. Note that the retrieved cloud pressure is close to the surface pressure in the cloud shadows, and the $NO_2$ retrieval for polluted conditions is very
sensitive to the cloud pressure retrieval. The cloud pressure differences between both cloud algorithms are usually less than 100hPa (not shown), and this leads to a change in AMF by more than a factor of 2.

Another possible extension of the cloud retrieval algorithm is to use a more realistic cloud treatment, such as the clouds-as-layers (CAL) approach, which treats the cloud as a uniform layer of light-scattering water droplets, instead of Lambertian cloud model. This approach has been used to investigate the $NO_2$ retrieval from GOME-2 based on the OCRA/ROCINN cloud
retrieval (Liu et al., 2020, 2021). However, OCRA/ROCINN uses a sophisticated approach (Loyola et al., 2018), and to develop such a cloud retrieval algorithm is beyond the purpose of this study. Instead, a simple approach is applied, which assumes that the cloudy scenes are 100% covered by a uniform layer of water cloud with a 1-km geometrical thickness. The cloud single scattering albedo is set as 1 and the asymmetry parameter is assumed to be 0.85. These values are consistent with those used in the cloud and $NO_2$ retrieval (Liu et al., 2020, 2021). The cloud correction in the $NO_2$ retrieval uses the same cloud properties
as the cloud retrieval. This approach retrieves cloud top pressure and optical thickness based on measured reflectances at 460

nm and $O_2$-$O_2$ SCD or three 1-nm (758–759 nm, 760–761 nm and 765–766 nm) averaged radiances around the $O_2$-A band. In addition, negative cloud optical thicknesses are allowed by using extrapolation in the retrieval in order to treat the cloud shadow situations.

Examples of CAL cloud retrievals are shown in Figure 12a and Figure 12c. In the cloudy scene, the cloud optical thickness retrieval is around 8, and the cloud top pressure is slightly higher than 700hPa, which is close to the true value. There is a small difference between the retrievals from the $O_2$-$O_2$ and $O_2$-A band. The bias of the $NO_2$ AMF using the CAL cloud is slightly smaller than the bias based on the Lambertian cloud correction. In the shadow, the retrieved cloud optical thicknesses are negative, and the value from the $O_2$-A retrieval is slightly smaller. The cloud top pressure is higher in the shadow than in the neighboring cloudy pixels. The bias of the $NO_2$ retrieval using the $O_2$-$O_2$ and $O_2$-A CAL clouds is slightly different from the bias using the Lambertian cloud correction. In general, there is a good agreement between the $NO_2$ retrievals using the CAL and Lambertian cloud models, and the biases of the retrieval corrected by the CAL cloud are slightly higher than those based on the Lambertian cloud correction (Figure 12b and Figure 12d).

### 4.1.2 $NO_2$ AMF using fitted surface albedo

In the cloud shadow, the standard $NO_2$ retrieval algorithm, which uses a known surface albedo, has a positive bias in the retrieved AMF, whereas the TOA reflectance shows a negative bias compared to the corresponding clear sky reflectance (Figure 13a). In an attempt to compensate for such a positive bias, we calculate the AMF using an effective surface albedo based on the measured reflectance. The surface is assumed to be a Lambertian reflector, and the surface albedo is obtained by fitting the simulated reflectance at TOA in a pure Rayleigh scattering atmosphere under a cloud-free condition. The retrieved albedo is then used for the $NO_2$ AMF retrieval. Figure 13a shows that the bias of the retrieval based on AMFs calculated using an effective albedo is significantly reduced in the cloud shadow. However the correction approach tends to increase the retrieval bias for clear sky pixels outside of the cloud shadow, and for the cloudy region, the retrieval bias based on the effective albedo is much larger than using the standard approach.

In order to verify the feasibility of the correction approach, we compare the biases of the $NO_2$ retrieval for the standard retrieval approach and calculations based on a fitted surface albedo for various box cloud scenarios, as shown in Figure 13b. As can be seen, the retrieval is improved for most of cases, however higher biases are found for high cloud cases (shown as stars in the figure). Further investigations indicate that the retrieved surface albedo is 0 for these pixels, which introduces a large negative error in the AMF calculation. It should be noted that the retrieved albedo value is restricted between 0 and 1. Therefore, the measured radiance for such pixels is smaller than or equal to the corresponding radiance with an albedo of 0 for clear sky condition. This correction can be extended to satellite measurements where the fitted surface albedo is lower than climatological values which may reduce retrieval errors due to surface albedo uncertainties. However, the surface albedo at the UV-visible wavelengths is usually small. Since the $NO_2$ AMF calculation is very sensitive to surface albedo, especially for low surface albedo and polluted regions (Boersma et al., 2004), Such cases can cause significant error in the $NO_2$ retrieval.

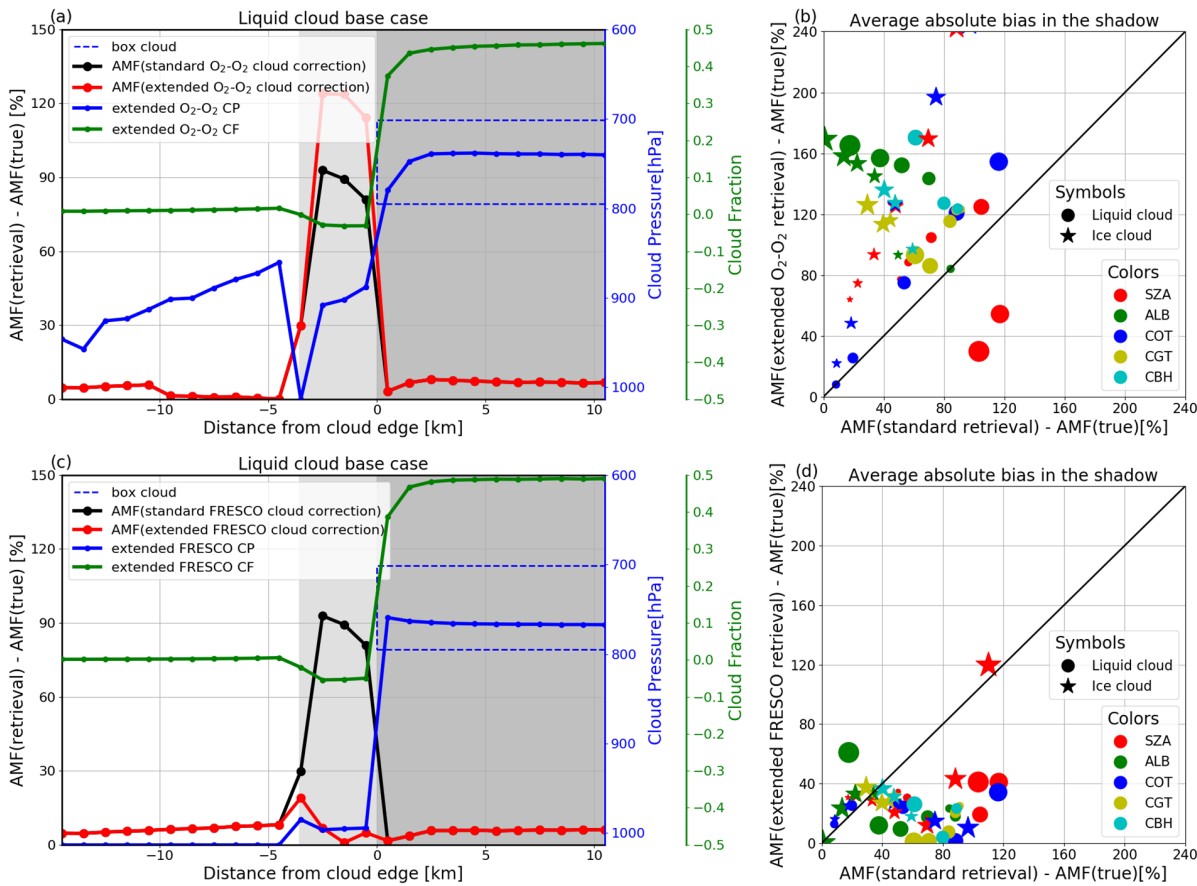

**Figure 11.** Examples of NO$_2$ AMF retrieval using the extended O$_2$-O$_2$ (top) and FRESCO (bottom) cloud. (a) and (c) Comparison of the AMF biases based on the standard retrieval approach and the AMF calculated with the extended cloud retrievals for liquid cloud base case. The dark gray, light gray and white regions represent cloudy, cloud shadow and clear scene, respectively. (b) and (d) Comparison of the AMF biases for the simulations with a box-cloud. Each point represents the average bias in the cloud shadow, and colors correspond to various parameters for the cases with the liquid cloud (circles) and ice cloud (stars). The biases are shown in relative value, and the various marker sizes represent different parameter values.

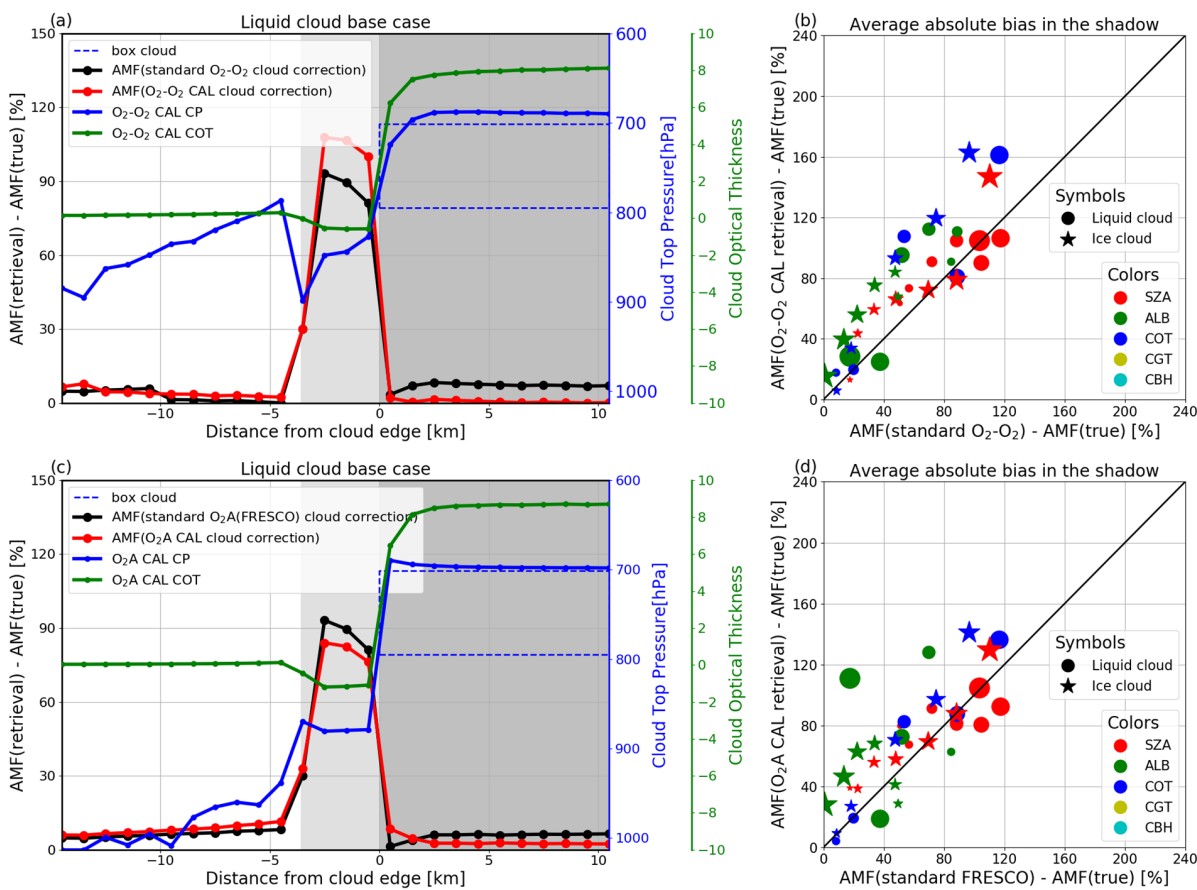

**Figure 12.** Similar to Figure 11, but the NO$_2$ retrieval uses cloud correction based on a clouds-as-layers approach. Cloud pressures given on the right of the figure are cloud top pressures. X-axis on the right of the figure represents the retrieval bias based on the NO$_2$ retrieval using the standard O$_2$-O$_2$ (b) and FRESCO (d) clouds, respectively.

### 4.1.3 AMF scaling by O$_2$-O$_2$ SCD

An alternative approach to correct the NO$_2$ retrieval in the cloud shadow is to use the difference between the retrieved O$_2$-O$_2$

SCDs and the reference calculations for a clear scene under the same condition:

$$M_{NO_2} = M_{NO_2}^{clr} \cdot (S_{O_2\text{-}O_2}^{meas} / S_{O_2\text{-}O_2}^{clr}) \tag{8}$$

where $M_{NO_2}^{clr}$ and $S_{O_2\text{-}O_2}^{clr}$ are the NO$_2$ AMF and the O$_2$-O$_2$ SCD calculated for the clear scene, and $S_{O_2\text{-}O_2}^{meas}$ is the O$_2$-O$_2$ SCD derived from the observed spectrum. In the cloud shadow regions, the retrieved cloud fraction is 0 since the measured reflectance is smaller than the corresponding clear sky reflectance. As a result, the AMF in the retrieval is the clear sky AMF.

The basic idea of this correction approach relies on the assumption that there is a certain degree of similarity between the O$_2$-O$_2$ and polluted NO$_2$ profiles, since both species have highest concentration near the surface. However since profiles are

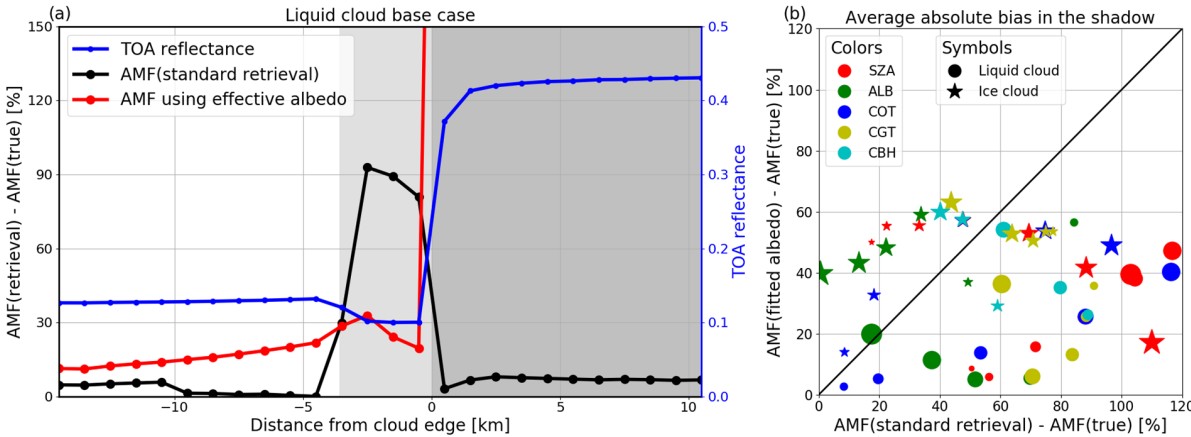

**Figure 13.** Similar to Figure 11, but for the NO$_2$ AMF using the effective surface albedo. Notice that the x- and y- range in (b) are [0, 120] instead of [0, 240].

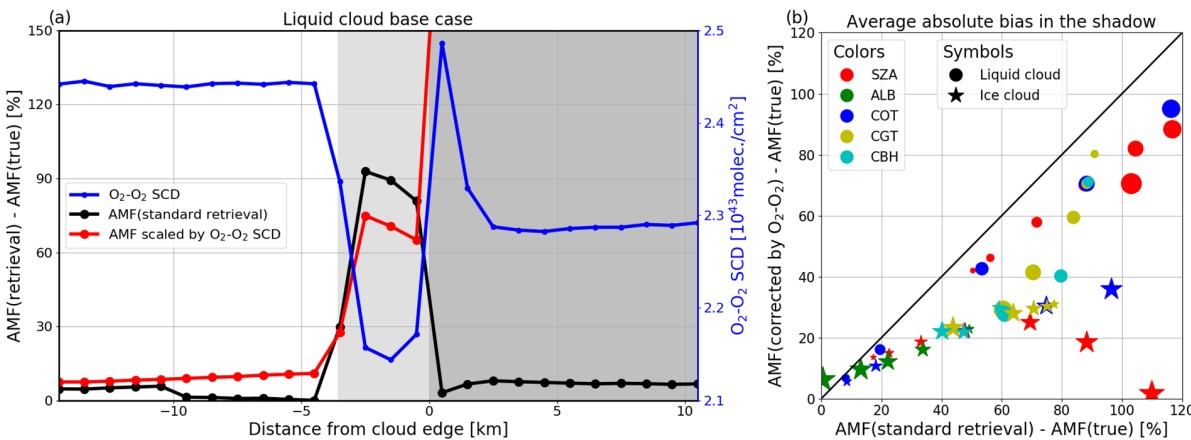

**Figure 14.** Similar to Fig. 13, but the AMF retrieval is corrected using a ratio of the retrieved O$_2$-O$_2$ SCD and the reference SCD calculated for a clear scene under the same condition.

not identical, the method can only partly correct for cloud shadow effects. Figure 12 shows a clear negative correlation between O$_2$-O$_2$ SCD and the standard retrieval bias. After applying the correction using Eq. 8, the biases are reduced by about 20% in the shadow. Again, this approach is not suitable for the cloudy pixels. For the synthetic box cloud cases, the retrieval bias is

465 systematically reduced when the correction approach is used. The improvement is 10-30% for the low cloud cases and is more noticeable for the high cloud cases.

### 4.1.4 Parameterization approach

Following the discussion in Section 3, the error of the $NO_2$ retrieval in the cloud shadow depends on the cloud shadow fraction, slant cloud optical thickness, $NO_2$ profile, neighboring pixel cloud top height, surface albedo, as well as the solar-satellite geometries. Ideally, the 3D bias can be quantified as a function of the above parameters and stored in the LUT. However, there is a limited number of synthetic datasets due to the limited computational resources. Based on the current dataset, an exercise can be made for the condition with a nadir view (VZA=0°) and a surface albedo of 0.05. In such conditions, the bias of the $NO_2$ retrieval due to cloud shadow effects can be described as:

$$\sigma_{NO_2} = F_1(PH) \cdot (1 - F_2(NCTH)) \cdot F_3(\log(SCOT)) \cdot (1 - CSF) \tag{9}$$

where $PH$ is the $NO_2$ profile height, $NCTH$ is the cloud top height of the neighboring pixel, $\log(SCOT)$ is the logarithm of slant cloud optical thickness, and $CSF$ is the cloud shadow fraction.

$F_1$, $F_2$, $F_3$ are all quadratic polynomials, and the coefficients of the polynomial are obtained by fitting the averaged $NO_2$ AMF bias in the cloud shadow from a series of simulations with a box-cloud as presented in Section 3. The cases with a cloud shadow area larger than 16 km are excluded from the analysis (e.g. SZA=80° for low cloud and SZA=70°, 80° or CBH=12 km for high cloud) since the synthetic data only simulates the spectra at 0-15 km away from the cloud edge. We obtain the following results:

$$F_1(x) = 0.75 - 0.17x + 0.01x^2 \tag{10}$$
$$F_2(x) = -0.42 - 4.32x + 0.34x^2 \tag{11}$$
$$F_3(x) = 0.01 - 0.15x + 0.30x^2 \tag{12}$$

As can be seen in Figure 15, the difference between the parameterization estimation and the true bias is mostly within 20%.

### 4.2 Comparison of mitigation strategies for synthetic data

We applied the correction approaches described in previous sections to $NO_2$ retrievals applied to synthetic dataset with realistic LES clouds. Figure 16 compares AMF biases obtained using the correction approaches described in Sections 4.1.1, 4.1.2 and 4.1.3 with retrieval biases from the standard algorithm.

For the first approach(Sections 4.1.1) based on the extended Lambertian cloud correction, only pixels with the extended $CF_r$<-1% are used. For such cases, the standard retrieval uses the clear AMF. The AMF bias based on the corrected approach is close to the standard bias when the $CF_w$ is close to 0, and the discrepancy increases for lower $CF_w$ values. In many cases, the bias of the retrieval using the extended $O_2$-$O_2$ correction is much larger than the bias from the standard retrieval (Figure 16a), while significant improvement can be obtained for retrievals using the extended FRESCO correction (Figure 16b). It should be noted that the wavelength dependency of the surface albedo is not considered in our analysis. In reality, there is a large difference in the surface albedo value between visible and near-infrared wavelengths for many regions (see (Emde et al., 2022)), and this may lead to different results for the retrieval using the FRESCO correction. For the correction based on the

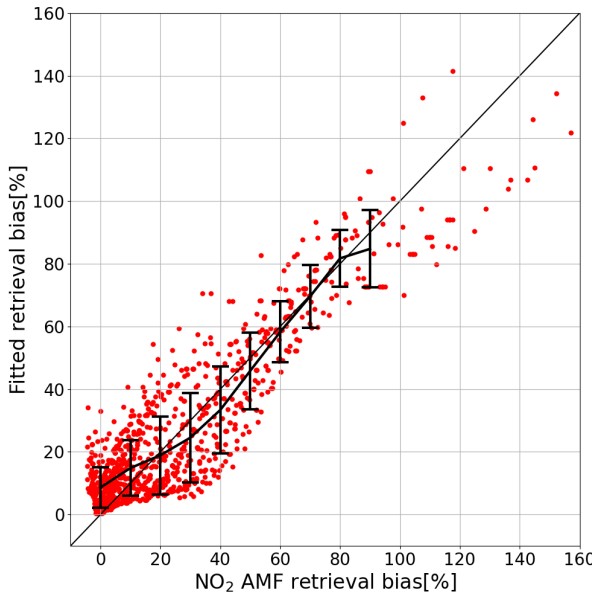

**Figure 15.** Comparison of the AMF bias in the cloud shadow based on the standard $NO_2$ retrieval algorithm and the estimated bias based on Eq. 9 for the box cloud cases.

CAL cloud, the pixels with the standard $CF_w$<50% are used. Results show that the biases for the corrected approach are similar to those using the standard Lambertian cloud correction (Figure 16c and Figure 16d), and this bias is slightly higher for negative cloud optical thickness pixels. In general, there is basically no improvement in comparison to the standard retrieval approach.

For the second and the third correction approaches(Sections 4.1.2 and 4.1.3), only pixels with $CF_w$<50% are used, and the standard retrieval uses cloud correction based on the $O_2$-$O_2$ cloud product. For cloud-free pixels ($CF_w$=0), both approaches can partly improve the retrieval due to the cloud shadow effects (Figure 16e and Figure 16f). When using effective surface albedos, biases are reduced by about 30%, while a 40% improvement is obtained when using AMFs corrected by a ratio of the measured $O_2$-$O_2$ SCD. However biases significantly increases when $CF_w$>0, especially when using effective albedos to correct AMFs (Figure 16e). In summary, improvements are obtained using both approaches, but they are limited to cloud-free pixels.

Figure 17 presents examples of the parameterization approach (Section 4.1.4) for the synthetic data. Since the approach investigated here is based on the analysis of a limited dataset, the dependency on observation geometry and surface albedo is not taken into account, therefore, we focus on scenarios with VZA of 0° and surface albedo of 0.05, consistent with conditions considered in Section 4.1.4. The first and second column in Figure 17 represent results for SZA of 40° and 60°, respectively. The first row (Figure 17a and b) shows the bias of the $NO_2$ AMF retrieval based on the standard retrieval approach, including a cloud correction based on the $O_2$-$O_2$ cloud retrieval. As usual, cloudy pixels ($CF_w$>50%) are excluded from the analysis. The bias in the clear sky region is generally less than 5%, except for the pixels next to clouds, which is probably due to cloud shadow effects. In order to obtain the parameters needed for the correction approach, the synthetic input of 3D fields of cloud content

from ICON is used, which includes 588×624 pixels for the full domain. Each simulated pixel includes 6×6 ICON cloud pixels. The SCOT is calculated for each subpixel using MYSTIC, and is averaged for the simulated pixel (Figure 17e and f). The pixels affected by 3D clouds need to meet those conditions: nearly cloud-free from the satellite view, but affected by the neighboring clouds shadows. Here, we use COT<3 (corresponding to $CF_w$<50% for the nadir view) to define nearly clear sky, and SCOT>1 (the $NO_2$ bias becomes significant for SCOT>1 as shown in Figure 10) to determine the pixels affected by cloud

shadows. The CSF is the ratio of the cloud shadow affected sub-pixels (in the simulated pixel) to the total number of subpixels. Results are shown in Figure 17c and d. The cloud top height (not shown) is the maximum value of 6×6 cloud pixels from the southern neighbor, which is from the direction of the Sun. Finally, the estimation of the bias is displayed in Figure 17g and h. Note that the estimated bias map has a similar pattern as the true bias. The scatter plots comparing estimated and true $NO_2$ biases for the cloud shadow affected pixel ($CF_w$>10%) are given in Figure 17i and j. Result shows a good general agreement,

however, some differences exist, since the real situation is complex and not necessarily well captured by approximations used in our approach. In particular, a $CF_w$ dependency can be found in the results. The true retrieval bias for the high $CF_w$ is smaller than the bias for the low $CF_w$ under the same condition. This is probably due to the simplified cloud correction approach. As discussed in Section 3.5, the total error is a linear weight of the error due to the 3D effect in the cloud shadow and the error from the simplified cloud correction for cloudy pixels. The latter is not included in the current parameterization approach.

## 4.3 Comparison of mitigation strategies for observed data

In order to investigate the impact of mitigation strategies discussed above on observed data, one needs to identify 3D cloud cases. For TROPOMI, we selected two cases (24 March 2019 and 30 December 2019) as discussed in **?**. The latter case is used to investigate the effect of the proposed mitigation strategies on real data. For this case, there is a clear cloud band, and a completely cloud-free scene with a large extent of a cloud shadow region in the North of the cloud (as shown in Figure 18). The

535 first correction approach (Section 4.1.1) is not applied to the TROPOMI data, since cloud fractions from the current TROPOMI cloud retrievals are confined to the intreval [0, 1] (Loyola et al., 2018; van Geffen et al., 2021) and building a new extended TROPOMI cloud retrieval is beyond the scope of this study.

First, our $NO_2$ AMF retrieval script is adapted to TROPOMI. The effective surface albedo is fitted at 437.5 nm, which is the wavelength used for the AMF calculation. The $O_2$-$O_2$ SCD retrieval follows Veefkind et al. (2016) and includes a

540 correction for its dependency on the temperature profile (Veefkind et al., 2016). The $NO_2$ retrieval using the standard approach, together with the two retrievals using our proposed correction methods (fitted surface albedo and $O_2$-$O_2$ SCD) are shown in Figure 19a. The three retrievals agree very well over the clear sky region (white region). In the cloud shadow, the $NO_2$ VCD using the correction approaches is larger than the corresponding $NO_2$ column from the standard retrieval. In order to validate the correction approaches, we compare the averaged $NO_2$ column in the cloud shadow and around the shadow as shown in

Figure 19b. The $NO_2$ around the shadow is the average of the $NO_2$ column using the standard approach for 4 pixels in the clear region and 4 pixels in the cloudy region. We assume that this represents the true $NO_2$ column. The standard $NO_2$ column in the cloud shadow is systematically lower than around the cloud shadow region due to the 3D cloud effects, and the differences are reduced when the retrieval includes the correction in the shadow. The AMF corrected by $O_2$-$O_2$ SCD improves the retrieval

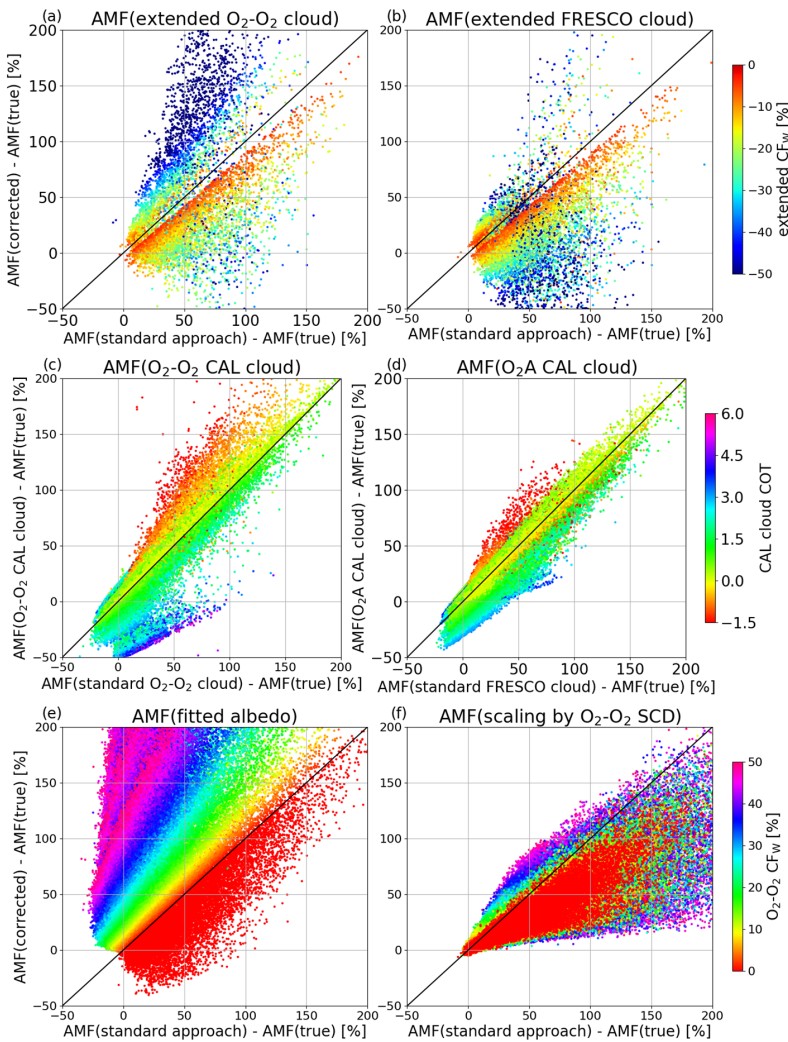

**Figure 16.** Comparison of the AMF bias using the standard retrieval algorithm and three correction approaches for the synthetic data with realistic LES clouds. (a)-(d) the AMF calculation based on the extended cloud retrievals, including extended standard $O_2$-$O_2$ (a) and FRESCO (b) cloud retrieval, and the CAL retrievals based on $O_2$-$O_2$ (c) / $O_2$-A (d) absorption. (e) the corrected AMF calculated using an effective surface albedo, and (f) the correction based on a scaling using $O_2$-$O_2$ SCDs. For (a) and (b), only pixels with the retrieved $CF_r$<1% are included in the analysis, and for (c)-(f), the pixels with the standard $CF_w$<50% are used. The colors represent the retrieved $CF_w$ (top and bottom) and COT (middle).

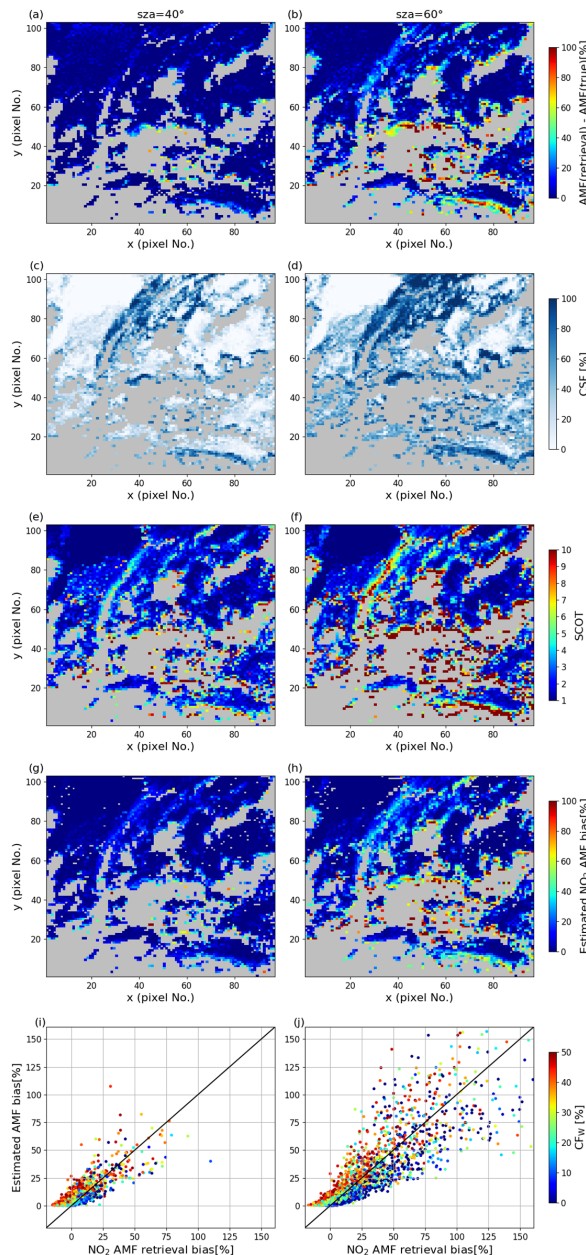

**Figure 17.** Example of our parameterization approach for the NO$_2$ retrieval bias in the cloud shadow for the LEO cases with surface albedo=0.05, VZA=0°, SZA=40° (left) and 60° (right). (a) and (b) show the bias of NO$_2$ retrieval based on the standard retrieval algorithm using O$_2$-O$_2$ cloud correction. Grey shaded pixels indicate cloudy pixels. (c) and (d) are the cloud shadow fraction. (e) and (f) are the averaged slant cloud optical thickness. (g) and (h) are the estimated NO$_2$ bias using the Eq. 9. (i) and (j) compares the true retrieval bias with the estimation, only the pixels with the cloud shadow fraction > 10% and slant cloud optical thickness > 1 are used in the analysis, the colors represent the cloud radiance fraction from the retrieval.

for all cases, while the AMF calculated by the effective surface albedo seems to overcorrect for rows 395 and 396. For these cases, the retrieved surface albedo for the pixels in the cloud shadow is 0 (lower limit), which is similar to the results that have been discussed in Section 4.1.2.

The parameterization approach relies on parameters, such as cloud shadow fraction, slant cloud optical thickness, the $NO_2$ profile and neighboring cloud top height. In practice, the $NO_2$ profile height is based on the $NO_2$ vertical profiles from the TM5-MP model (van Geffen et al., 2021), which is used for the calculation of the AMF in the operational product. The cloud top height is a maximum of VIIRS cloud height for the neighboring pixels of the TROPOMI pixel. The COT and cloud shadow mask is not available for VIIRS data for this case, probably due to the large SZA ($\approx 80°$). Therefore we use an alternative approach based on the correlation of COT and $CF_r$ from the 1D simulations described in Section 2.5, taking advantage of the fact that the $CF_r$ depends strongly on the COT and much less on the surface albedo and the solar and viewing geometries. The SCOT is computed using the SZA of the selected TROPOMI pixel and an averaged COT calculated over five neighboring TROPOMI pixels. Since the VIIRS CTH is up to 7 km, the cloud shadow area is about 40 km, which corresponds to 4.5 TROPOMI pixels. The cloud shadow fraction is based on the VIIRS M3 band. The averaged VIIRS reflectance over the clear pixel near the cloud edge is used as a reference to define whether the VIIRS pixels are in the cloud shadow, and then the cloud shadow fraction is computed. The averaged parameters over the shadow are shown in Figure 20a.

Finally, we estimate the $NO_2$ VCD bias using Eq. 9 for TROPOMI pixels located in the cloud shadow, weighted by the $NO_2$ VCD from the standard retrieval. In Figure 20b, the averaged $NO_2$ bias from the parameterization approach in the cloud shadow is compared with the difference of the $NO_2$ retrieval around and in the cloud shadow, each point represents the analysis for one row. Although there are only a few data points, the estimated bias shows a positive correlation with the $NO_2$ bias by comparing $NO_2$ retrieval in and around the shadow. The estimated value is however slightly larger. Besides the error due to the parameterization approach itself, the error from deriving various parameters from the satellite images may lead to uncertainties. Doubling the $NO_2$ profile height or halving the slant cloud optical thickness lead to a reduction of the bias by 3% or 13% respectively (see Figure 20b). Although it is error prone due to the complexity of the problem and the difficulty to extract relevant parameters from imager data, the parameterization approach might be very useful to identify satellite pixels likely affected by significant 3D clouds biases.

It should also be noted that other sources of uncertainty in the $NO_2$ retrieval itself may affect such comparison results, in particular the uncertainty on parameters used in the AMF calculation, e.g. the a priori $NO_2$ profile shape. For high SZAs, uncertainties due to the slant column retrieval from the spectral fit and the stratospheric correction are also important. In addition, the true $NO_2$ column is unknown, and $NO_2$ columns usually show a considerable spatial variability, especially over polluted regions. Therefore, without additional independent measurements, the 3D effects on $NO_2$ retrievals are difficult to identify and correction approaches are hard to validate.

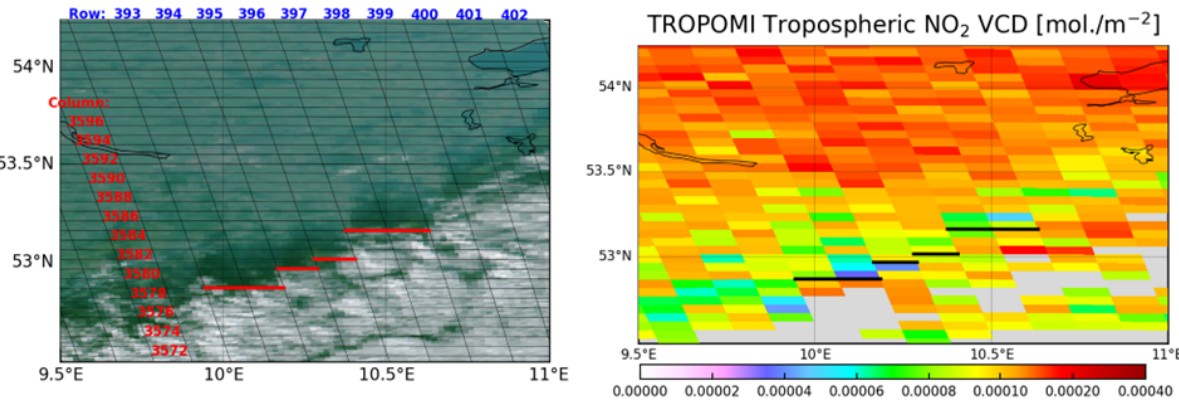

**Figure 18.** Example of satellite observation for the cloud shadow band on 30 December 2019. Left panel: the VIIRS RGB image with TROPOMI footprint. Right panel: the TROPOMI tropospehric $NO_2$ VCDs, the gray regions represent pixels with $CF_w > 50\%$. The red(left)/black(right) lines indicate the cloud edge in along-track direction from row 393 to 398.

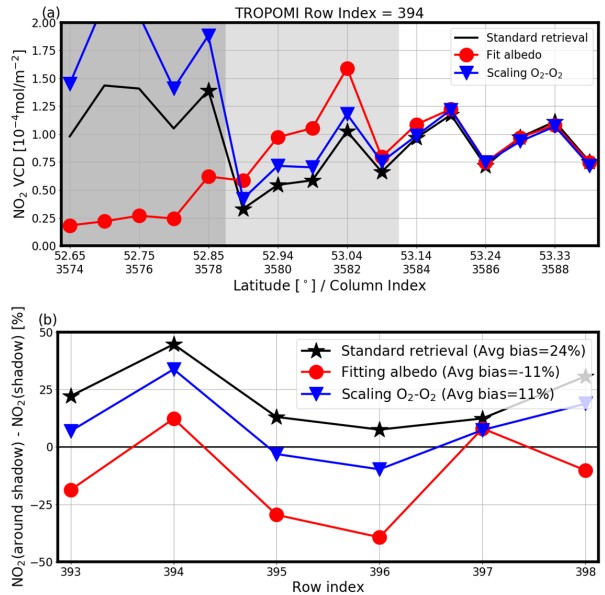

**Figure 19.** Comparison of the $NO_2$ VCDs using a standard retrieval algorithm and retrievals implementing the correction approaches discussed in Sections 4.1.2 and 4.1.3. The data use TROPOMI measurements over the cloud shadow band for 30 December 2019. Top panel: the $NO_2$ retrieval based on three approaches as a function of latitude for TROPOMI row 394. The dark gray, light gray and white regions represent the cloudy, shadow and clean regions, respectively. Bottom panel: difference of the $NO_2$ columns in the cloud shadow and that around shadow for the standard retrieval and the retrieval including a correction in the cloud shadow for row 393-398, and the average bias over all rows is given in the legend. See text for further details.

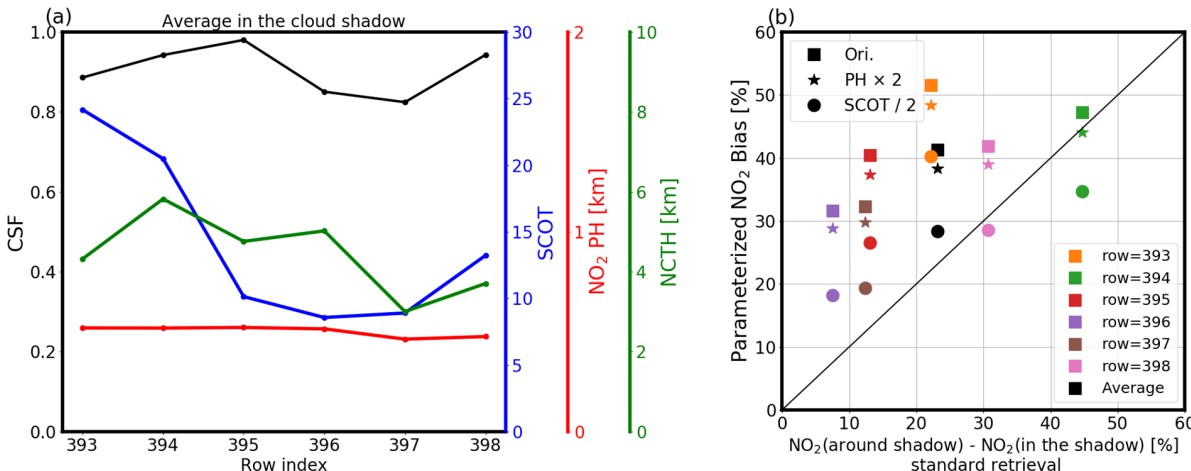

**Figure 20.** Estimation of NO$_2$ retrieval biases over the cloud shadow bands from TROPOMI measurements on December 30, 2019. (a) Averaged parameters in the cloud shadow, which are used to estimate the bias. (b) Comparison of the estimated bias and the NO$_2$ bias calculated based on the difference of NO$_2$ retrieval around and in the cloud shadow. The black is the average over all the rows, the stars and the circles correspond to the estimation using doubled NO$_2$ profile height and halved slant cloud optical thickness. See the text for further details.

## 5  Conclusions and Outlook

In this study, we have investigated the impact of 3D clouds on the tropospheric NO$_2$ retrieval from UV-Visible sensors. In order to identify and quantify this impact, we first applied standard NO$_2$ retrieval methods including cloud corrections to synthetic data generated by the 3D Monte Carlo radiative transfer model MYSTIC. Since the cloud correction schemes are based on a simple cloud model, the accuracy of the NO$_2$ retrieval depends on not only the cloud retrieval, but also on other factors, such as the NO$_2$ profile. The analysis in the study focused mainly on the error of the NO$_2$ retrieval due to the 3D cloud effects. Then, a sensitivity study for the simulations including a box-cloud was made, and dependencies on various parameters were investigated. Finally, possible mitigation strategies such as AMF correction methods, and a parameterization approach were investigated and compared based on realistic simulations with LES clouds and observed data.

The most significant biases are related to cloud shadow effects. The cloud products used in the NO$_2$ retrieval treat the cloud shadow pixels as cloud-free, resulting in large positive biases (up to more than 100%) in the NO$_2$ AMF calculation. The magnitude of cloud shadow effects depends on the NO$_2$ profile and is larger for polluted profiles, i.e. for profiles containing significant NO$_2$ amounts in the lower troposphere. The retrieval bias depends strongly on the cloud shadow fraction, and we find that pixels affected by 3D cloud effects can be corrected using an independent pixel approximation, which assumes that the retrieval bias can be written as a linear combination of the bias from the clear, cloud shadow and cloudy parts. If the cloud shadow area is smaller than the size of the satellite pixel, the cloud shadow effect will be significantly reduced. We conclude

that cloud shadow fraction, $NO_2$ profile, cloud optical thickness, solar zenith angle, as well as surface albedo are the most important parameters to characterize 3D cloud impacts on $NO_2$ retrievals.

Several approaches to correct the $NO_2$ retrieval in the cloud shadow were explored based on both synthetic and observational data. These includes: (a) the AMF retrieval using cloud correction based on the extended $O_2$-$O_2$/FRESCO and CAL cloud retrievals. (b) calculation of the AMF using an effective surface albedo based on the measured radiance. (c) correction of the $NO_2$ retrieval by using the difference of retrieved $O_2$-$O_2$ SCDs and reference calculations for a clear scene under the same geometry. The latter two methods can partly correct the cloud shadow effects in the $NO_2$ retrievals. However, they are limited to cloud-free conditions. Furthermore, an approach was developed to identify in real data the $NO_2$ measurements that are likely biased due to 3D cloud effects. The approach estimates the size of the $NO_2$ bias using an empirical formula based on relationships derived from an analysis of model simulations. It provides a way to improve the current data flagging method.

In future work, the development of improved parameterization approach accounting for 3D cloud effects requires appropriate and extended synthetic dataset covering a large range of atmospheric situations. Since 3D cloud effects depend in a non-trivial way on many parameters, Machine Learning approaches may provide a fruitful way for development of parameterization mitigation methods of 3D cloud impacts on UV and visible trace gas products. Another possible mitigation method is to develop more sophisticated cloud retrievals, which account for the 3D effects, are feasible to apply to satellite observation, and can easily adapt to current trace gas retrieval algorithms.

Moreover, the validation of the mitigation methods is needed. Such validation is non-trivial and possibly requires new experimental approaches for measurements of both cloud shape and trace gas spatial variation. For example: for cloud shadow effect estimation a cloud shadow product is needed. 3D radiative transfer simulations as those utilized in this study, but for all relevant spectral bands, may be used to test and validate such algorithms. However, a complete validation must include comparison with independent measurements.

*Code and data availability.* The QDOAS software for DOAS retrieval of trace gases is available from https://uv-vis.aeronomie.be/software/QDOAS/. VIIRS data were accessed through the NOAA Comprehensive Large Array-Data Stewardship System (CLASS, https://www.bou.class.noaa.gov). TROPOMI data were downloaded from https://s5phub.copernicus.eu/.

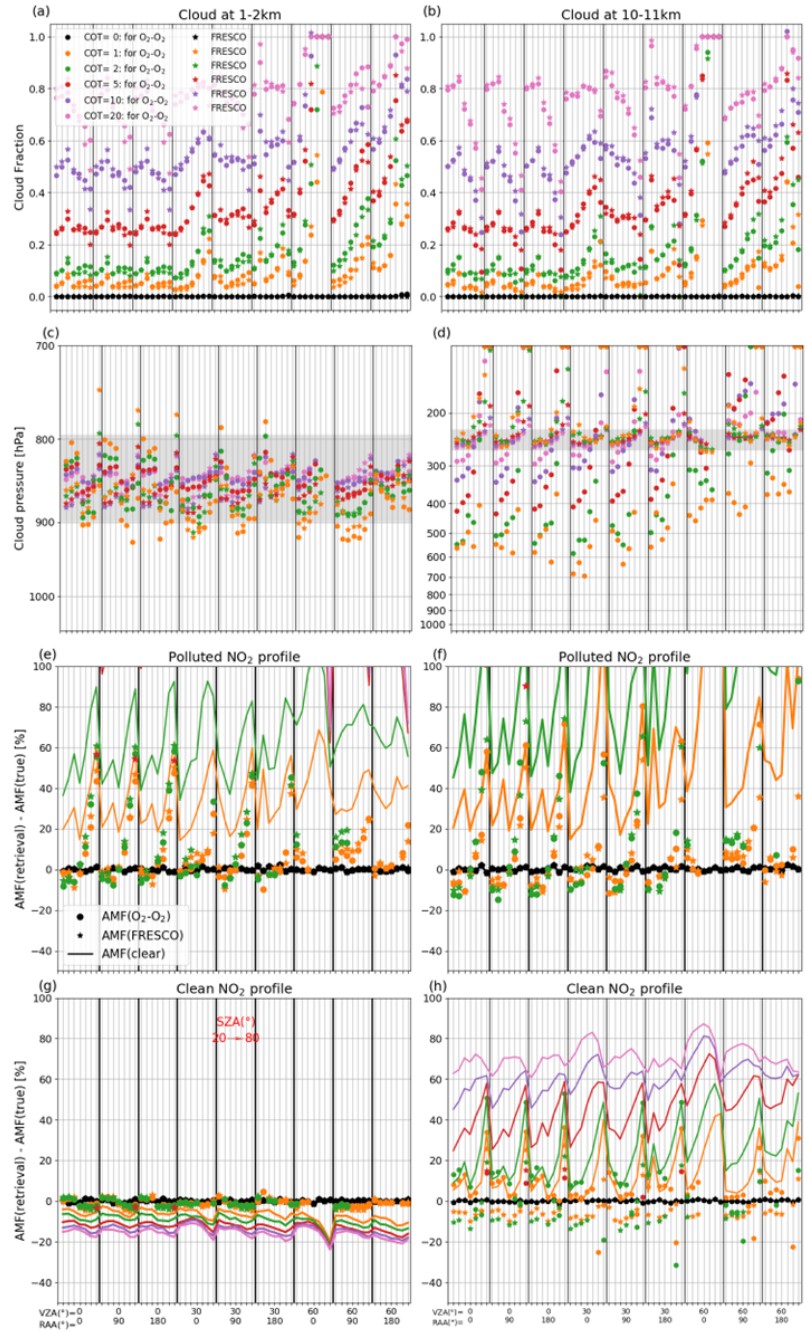

**Figure A1.** Examples of cloud and NO₂ retrieval for 1D cloud scenes, discussed in Section 2.5, with 1-2 km (left) and 10-11 km (right) cloud height. (a) and (b) show $O_2$-$O_2$ and FRESCO cloud fraction retrievals, (c) and (d) are the cloud pressure retrieval from $O_2$-$O_2$ and FRESCO cloud algorithms, the grey regions indicate the true cloud layer. (e)-(h) compare the bias of the NO₂ AMFs using cloud correction based on $O_2$-$O_2$ and FRESCO cloud products, as well as the AMFs without cloud correction, for polluted (e)/(f) and clean (g)/(h) condition. The cloud correction is applied when the pixels with CF$_w$ less than 50%. The x-axis represents the cases with different geometries. A variety of colors represent the cases with different cloud optical thickness.

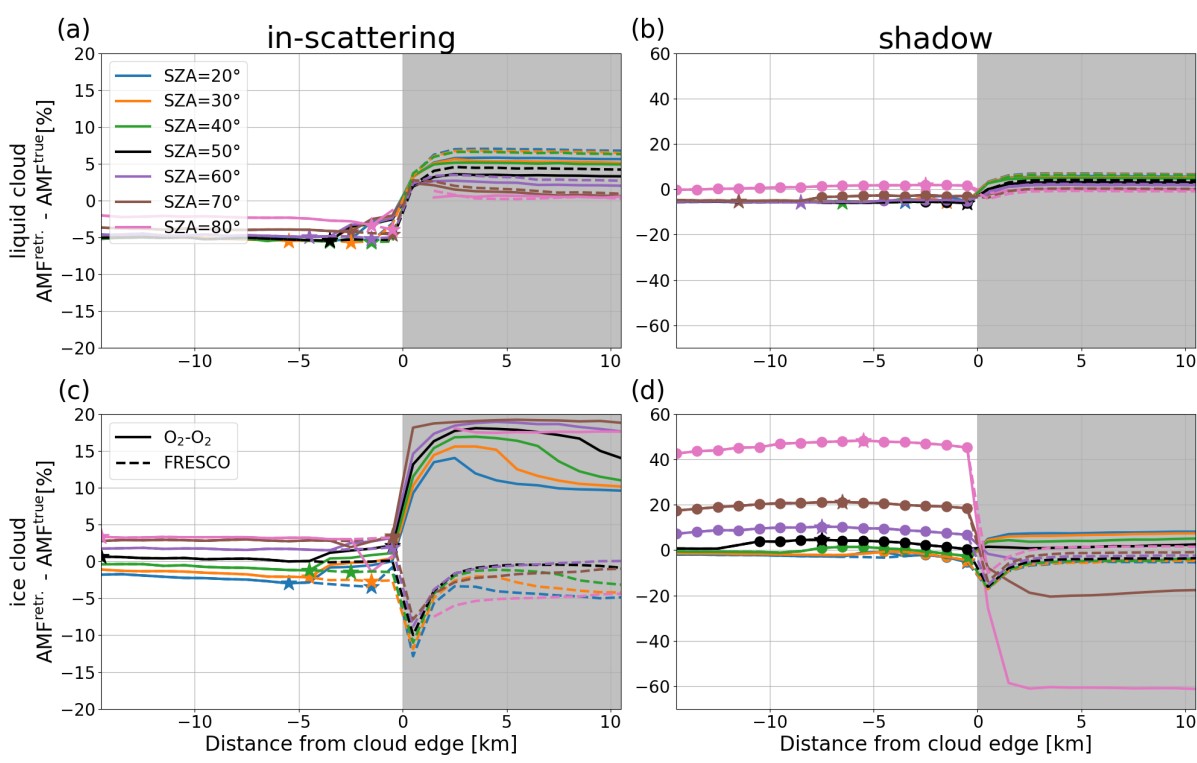

**Figure A2.** Similar to Figure 3, but the AMF retrieval using the clean NO$_2$ profile.

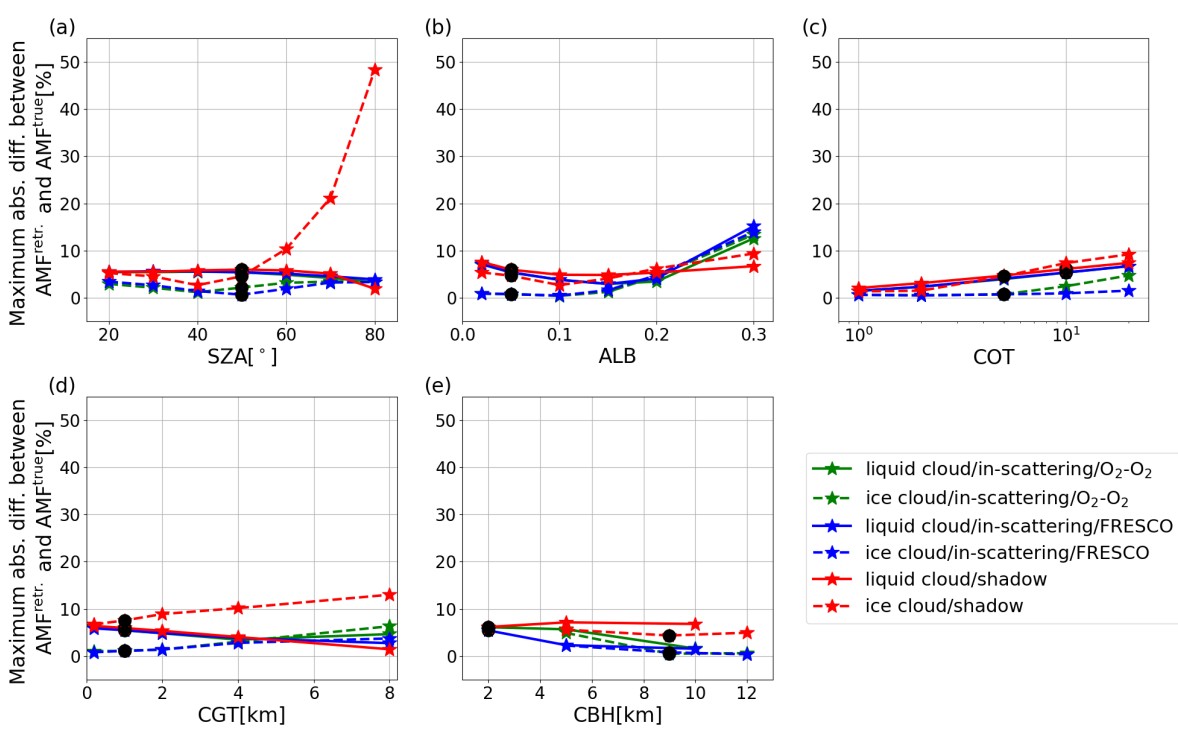

**Figure A3.** Similar to Figure 4, but the AMF retrieval using the clean NO₂ profile.

*Author contributions.* HY is the main contributor to the study. He applied the $NO_2$ retrieval algorithm on the synthetic data, analyzed the impact of 3D cloud on the retrieval, investigated the possible mitigation strategies, and he led the writing of this paper. CE provided synthetic data from 3D radiative transfer simulations. AK contributed to data analysis and software development. MvR, KS, BV and BM contributed to conceptualization and methodology. All co-authors have been involved into the discussion of results and the writing of this article

*Competing interests.* The authors declare that no competing interests are present.

*Acknowledgements.* This research activity has been funded by ESA (3DCATS project 4000124890/18/NL/FF/gp).

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
