# Peer review of "Impact of 3D Cloud Structures on the Atmospheric Trace Gas Products from UV-VIS Sounders - Part II: impact on NO2 retrieval and mitigation strategies"

_Atmospheric Measurement Techniques, 2021_

## Author Comment (AC1)

**Response to interactive comments from Referee #1**

We gratefully thank the reviewer for the careful reading of our manuscript and for the very constructive comments. Below the reviewer's comments are given in italic bold font. Our responses to the comments and how the comments have been addressed in the manuscript are shown in roman font.

***It would be good to have a short discussion of what the expected effect of aerosols is on the discussed 3d effects which here are discussed in a Rayleigh atmosphere.***

Without cloud, the effect of aerosol on trace gas retrieval is already complex. Thus, it hard to give the effect of aerosols on the 3D cloud effects, and it's rarely discussed in the literature. I have added a discussion in the introduction:

"The impact of aerosol on the trace gas retrieval is quite complex, which depends on many factors (Leitão et al., 2010), and it will become more complicated when 3D clouds are included. On the other hands, the effects of aerosols are very similar to the considerations made for clouds, and the aerosols are treated as clouds in some studies (Chimot et al., 2016). In this work, aerosols are not included."

***Page 2, last line: This sentence is a bit unclear as spatial heterogeneity will also be relevant in clear sky scenes and several effects are addressed at the same time here. Please separate into two (or more) sentences.***

The sentence has been rephrased to:

"In current atmospheric trace gas retrieval schemes from space sensors, clouds are treated in a simplistic way ignoring 3D structures and cloud shadows. The impact of 3D features like spatial heterogeneity and structured cloud boundaries increase when the spatial resolution of the instruments approaches the dimensions of cloud features."

***Page 3, line 14 / 15: It would be nice to have a very brief indication also of what Várnai et al. found in their work.***

We have included the following sentence: "the results indicate that the 3D radiative processes contribute to near-cloud reflectance enhancements, especially within 1 km from clouds."

***Page 9, line 21: I think it would be good to iterate here that only one aspect of possible errors introduced by cloud correction is covered. Perfect knowledge of all parameters is assumed and in particular, the $NO_2$ profile is assumed to be the same inside and outside of the cloud.***

We have included: "In this study, the calculation of NO2 AMF uses the perfect knowledge of all parameters, and in particular, the $NO_2$ profile is assumed to be the same inside and outside of the cloud. The only source of the error in the NO2 retrieval is introduced by cloud correction."

*Figure 2: I think that this display is somewhat misleading – I was tempted to see points close to the 1:1 line as "good" points while in reality, they are just points for which both cloud retrievals perform similarly. The main point of the discussion here is how large errors are and I think histograms of relative errors would be more appropriate.*

The figure is to show not only the bias the NO2 AMF retrieval due to the simplified cloud correction, and also comparison of the bias using different cloud products. It's difficult to display the latter when we use the histograms of the errors. We have added a group of figures in the appendix to show the examples of cloud and NO2 AMF retrieval for 1D clouds.

*Figure 10: It would be nice to have the same x-axis in both plots to allow direct comparison*

Correction made as suggested.

*Section 4.1.1 It would be interesting to add a short discussion of what you think about the surface albedo fitting implemented in the current TROPOMI lv2 product where the surface albedo is determined from radiance in case it is lower than the climatological value for a scene.*

We have made a statement:

"This correction can extend to the satellite measurements where the fitted surface albedo from the radiance is lower than the climatological value, and this may reduce the retrieval error due to surface albedo in certain situations. However, surface albedo at the UV-visible band is usually small. The $NO_2$ AMF calculation is very sensitive to surface albedo, especially for low surface albedo and polluted regions(Boersma et al., 2004). Such as the cases mentioned above cause significant error in the $NO_2$ retrieval."

*Cases where the retrieved albedo is 0 appear to be problematic – can you discuss this a bit more? Is that because the atmosphere is illuminated less than it would without cloud which reduces the backscattered intensity but does not change the layer AMF in the same way as a small albedo?*

In clear scene, the satellite measured radiance is the sum of backscattered radiance from the atmosphere and reflected radiance from the Earth's surface. Thus, we give an explanation:

"This means that the cloud leads to less photons through cloud into the shadow and back to the satellite, and this reduction is larger than the reflected radiance from the Earth's surface in corresponding clear scene."

*The application to TROPOMI data is based on the assumption that NO₂ retrievals should yield the same column in cloudy and clear regions as well as in the cloud shadow. However, considering the reduced actinic flux in the cloud shadow (and the increased values inside the cloud), shouldn't we actually see differences?*

This question is related to the impact of horizontal variation of the NO2 concentration, and this can be checked with the 3D box-AMF.

In general, the 3D effects will be larger/smaller when NO2 in cloud regions is higher/lower than NO2 in clear regions compared to the 3D effects for NO2 in cloud regions = NO2 in clear regions. On the other hand, the spatial scale of cloud shadow is comparable to the size of the TROPOMI pixels, and this effect may be small. This requires further investigation.

**Bibliography**

Boersma, K. F., Eskes, H. J., and Brinksma, E. J.: Error analysis for tropospheric NO2 retrieval from space, 109, https://doi.org/10.1029/2003jd003962, 2004.

Chimot, J., Vlemmix, T., Veefkind, J. P., de Haan, J. F., and Levelt, P. F.: Impact of aerosols on the OMI tropospheric NO2 retrievals over industrialized regions: How accurate is the aerosol correction of cloud-free scenes via a simple cloud model?, 9, https://doi.org/10.5194/amt-9-359-2016, 2016.

Leitão, J., Richter, A., Vrekoussis, M., Kokhanovsky, A., Zhang, Q. J., Beekmann, M., and Burrows, J. P.: On the improvement of NO2 satellite retrievals - Aerosol impact on the airmass factors, Atmospheric Measurement Techniques, 3, 475–493, https://doi.org/10.5194/amt-3-475-2010, 2010.

---

## Author Comment (AC2)

**Response to interactive comments from Referee #2**

We gratefully thank the reviewer for the careful reading of our manuscript and for the very constructive comments. Below the reviewer's comments are given in italic bold font. Our responses to the comments and how the comments have been addressed in the manuscript are shown in roman font.

***The quantities directly affected by 3d cloud effects would be the retrieved cloud fraction and cloud height.***

***These quantities are generally used for calculating $NO_2$ AMFs, and, as far as I understand, this should not be changed according to the authors.***

***But then it is essential to first check how far the cloud retrievals are affected by 3D effects before analysing the effects on trace gases.***

***For instance, a cloud shadow causes lower reflectance. This might actually be dealt with in the existing algorithms if negative cloud fractions would be allowed. This way it might be actually quite simple to account for cloud shadow effects without introducing new concepts/quantities like CSF.***

***Also other 3d effects (clouds in neighboring pixels) will affect the cloud fraction and cloud height retrieved based on IPA. It would be interesting to see to which extent these "wrong" CF/CH parameters do the $NO_2$ AMF correction intrinsically (such as aerosol effects being partly accounted for by the cloud algorithms yielding higher CF and lower CH than "reality").***

In this study, the 3D effects of $NO_2$ retrieval are discussed based on the classic $NO_2$ retrieval approach, which applied the cloud correction to the AMF calculation only for partly cloudy scene (the retrieved cloud fraction is larger than 0), otherwise, the scene is treated as cloud-free. The approach mentioned in the reviewer's comment, which uses the unrealistic cloud properties (negative CF), is not the standard approach. In addition, the cloud fractions are confined to [0,1] in the current TROPOMI cloud products (Loyola et al., 2018; van Geffen et al., 2021).

On the other hand, this approach can be added in the "Mitigation" part, which is one of possible way to improve the current $NO_2$ AMF calculation in the cloud shadow, called "AMF using extended cloud retrievals".

***I would thus like the authors to add an analysis of 3D effects on the cloud products first. The further mitigation strategy might be different if 3D effects could already be accounted for by e.g. negative cloud fractions. In any case, the mitigation strategies cannot be discussed without knowledge on the effect of 3D cloud structures on the standard cloud products themselves.***

We do not agree to add "an analysis of 3D effects on cloud products", since the main focus in the study is analysing the 3D effects on the $NO_2$ retrieval, and the cloud products used for cloud correction in the $NO_2$ retrieval are based on a simple cloud mode and obtain the effective cloud properties (CF, CH). The accuracy of cloud retrieval does not link to the accuracy of cloud correction, especially for the nearly cloud free scene, which is the main concern for the $NO_2$ retrieval.

We add a series of Figures (Figure 1) in Appendix to give the examples of cloud and $NO_2$ retrieval for 1D cloud cases, which show that the FRESCO retrieval usually is closer to the true cloud height, but the $NO_2$ AMFs using the $O_2$-$O_2$ cloud correction often show better agreement with the true AMF, especially for the high cloud cases. Thus, we believe that the analysis of 3D effects on cloud products is not a relevant topic in this paper. In addition, an example of extended cloud retrievals in the cloud shadow is included in the section "AMF using extended cloud retrievals".

*Minor comments:*

*Page 1, Line 2: "generally implement Lambertian cloud models": This is not true, see for instance OCRA/ROCINN.*
The sentence has been rephrased to: "generally implement a simple cloud model"

*Page 1, Line 3: "photon path length corrections": to my understanding, the cloud algorithms interpret the measured O2 or O4 absorption in terms of a cloud height. This should be stated here.*
This has been stated after:

The latter relies on measurements of the oxygen collision pair ($O_2$-$O_2$) absorption at 477 nm or on the oxygen A-band around 760 nm to determine an effective cloud height.

*Page 2, line 6: "amount of the trace gas along the average path": this sounds like the average path could be calculated and then linked to the amount of trace gas. It is rather the average absorption along light paths.*
This has been rephrased to: "the integrated trace gas concentration along the light path"

*Page 2, line 19: "A simplified Lambertian cloud model is generally used": This is not true, see for instance OCRA/ROCINN.*
The sentence has been rephrased to:

"A simple cloud model is generally used, which treats cloud as a Lambertian surface or a scattering layer, relying on the concepts of cloud fraction, cloud top albedo and cloud top pressure(Acarreta et al., 2004; Loyola et al., 2018; Wang et al., 2008)."

[Figure]

Figure 1: Examples of cloud and NO$_2$ retrieval for 1D cloud scenes, discussed in Section 2.5, with 1-2 km (left) and 10-11 km (right) cloud height. (a) and (b) show O$_2$-O$_2$ and FRESCO cloud fraction retrievals, (c) and (d) are the cloud pressure retrieval from O$_2$-O$_2$ and FRESCO cloud algorithms, the grey regions indicate the true cloud layer. (e)-(h) compare the bias of the NO$_2$ AMF retrievals using cloud correction based on O$_2$-O$_2$ and FRESCO cloud products, as well as the retrieval without cloud correction, for polluted (e)/(f) and clean (g)/(h) condition. The cloud correction is applied when the pixels with CF$_w$ less than 50%. The x-axis represents the cases with different geometries. A variety of colors represent the cases with different cloud optical thickness.

**Bibliography**

Acarreta, J. R., de Haan, J. F., & Stammes, P. (2004). Cloud pressure retrieval using the O2–O2 absorption band at 477 nm. *Journal of Geophysical Research: Atmospheres*, *109*(5). https://doi.org/10.1029/2003jd003915

Loyola, D. G., García, S. G., Lutz, R., Argyrouli, A., Romahn, F., Spurr, R. J. D., Pedergnana, M., Doicu, A., García, V. M., & Schüssler, O. (2018). The operational cloud retrieval algorithms from TROPOMI on board Sentinel-5 Precursor. *Atmospheric Measurement Techniques*, *11*(1). https://doi.org/10.5194/amt-11-409-2018

van Geffen, J. H. G. M., Eskes, H. J., Boersma, K. F., Maasakkers, J. D., & Veefkind, J. P. (2021). *TROPOMI ATBD of the total and tropospheric NO2 data products, Report S5P-KNMI-L2-0005-RP, version 2.2.0.* https://sentinel.esa.int/documents/247904/2476257/Sentinel-5P-TROPOMI-ATBD-NO2-data-products

Wang, P., Stammes, P., van der A, R., Pinardi, G., & van Roozendael, M. (2008). FRESCO+: An improved O2 A-band cloud retrieval algorithm for tropospheric trace gas retrievals. *Atmospheric Chemistry and Physics*, *8*(21), 6565–6576. https://doi.org/10.5194/acp-8-6565-2008

---

## Author Response (AR2)

We gratefully thank the reviewer for the careful reading of our revised manuscript and for the constructive comments. Below the reviewer's comments are reproduced in italic bold font. Our responses to these comments are given in roman font and pieces of text added to the manuscript are displayed in blue font.

**Referee #1:**

*The authors have addressed many of the points raised and have added relevant additional evaluation for clouds as layers which is nice.*

*Two of my comments appear to have been misunderstood:*

*"It would be good to have a short discussion of what the expected effect of aerosols is on the discussed 3d effects which here are discussed in a Rayleigh atmosphere."*

*The authors reacted by briefly discussing the relevance of aerosols in NO2 retrievals, but my intention was to reflect in how far the assumption of a Rayleigh atmosphere may have an impact on their results. For example, scattering aerosols in a cloud shadow may have different impacts on NO2 AMF than scattering aerosols outside the cloud shadow. Maybe the authors can give their thoughts on how relevant the neglection of aerosols is for their results.*

This discussion on the impact of aerosols has been removed from the introduction and rewritten under section '2.2 cloud correction':

"Aerosols are not included in this study. However, the presence of aerosol may lead to different impacts on the 3D effects, depending on aerosol properties, such as single scattering albedo, optical depth and vertical distribution. For example, scattering aerosols in the cloud shadow will increase the AMF and compensate the shadowing effect, whereas strong absorbing aerosols may decrease the AMF and enhance the 3D effect. The resulting effect may be rather complex, and further investigation would be needed for an accurate evaluation of such effects. In addition, it should be noted that, in practice, aerosols are implicitly treated as clouds in actual retrievals since the effects of aerosols are expected to be similar to those of clouds (Boersma et al., 2004, 2011)."

*"The application to TROPOMI data is based on the assumption that NO2 retrievals should yield the same column in cloudy and clear regions as well as in the cloud shadow. However, considering the reduced actinic flux in the cloud shadow (and the increased values inside the cloud), shouldn't we actually see differences?"*

*The authors react with a reference to 3d-AMFs but my point here is, that it is not trivial to assign low NO2 in the cloud shadow to a retrieval problem as less NO2 is expected in non-illuminated regions. I think this fact should be mentioned in the discussion in section 4.3*

It is correct that we cannot rule out that photochemical effects take place in the cloud shadow/cloudy regions and locally modify the $NO_2$ concentration. However, a reduction in actinic flux would probably lead to an increase of the $NO_2$ concentration due to a shift in the $NO$-to-$NO_2$ ratio. In principle, the independent pixel approximation approach in the AMF calculation may account for this by using intensity weighted AMFs from cloudy, clear and cloud shadow parts. In reality it is however not possible to treat such effects in a rigorous way, since the overall impact of a change in illumination on the NOx chemistry cannot be quantified in a simple way. Spatial variability (due to chemistry or transport) is certainly important and responsible for significant uncertainties in the retrieval. In this study, TROPOMI observations were selected so that they can be divided into complete cloudy, clear and cloud shadow conditions. The uncertainty of the $NO_2$ retrieval is discussed in the last paragraph.

In order to better discuss the importance of spatial variability, I added to following statement in section '3.4: Change of spatial resolution':

"Note that the synthetic data used in this study assumes that the NO$_2$ column is the same in clear and cloudy regions as well as in cloud shadow. Consequently, the NO$_2$ retrieval is based on the same assumption. In reality, however, the NO$_2$ column usually shows significant to large horizontal variability, which leads to uncertainty in the retrieval. The importance of such effects cannot be easily assessed using tools available for this study, and would need to be further investigated."

*I also note that there are a few typos in the newly added text and that some figure references now are no longer correct.*

Done

**Referee #2:**

*I still see an inconsistency between the sophisticated investigation of 3d effects on NO2 AMFs on the one hand, and at the same time treating cloud effects following a simple independent pixel approximation.*

This study aims to investigate the impact of 3D cloud structures on the trace gas retrieval products from UV-Visible sensors. The current operational trace gas retrievals are based on the DOAS approach, which consists two steps: DOAS spectral fitting methods to get SCD, and conversion of SCD into VCD by means of calculated AMFs. The 3D effects mainly affect the calculation of the AMF, and the cloud correction in AMF is based on the independent pixel approximation. This approach could not capture all 3D effects, and this study is to identify which situation will produce significant bias due to 3D effects, and investigate possible (simple) mitigation strategies for such cases.

*I don't understand at all how the authors can state that "The accuracy of cloud retrieval does not link to the accuracy of cloud correction" (reply to Referee #2), as clouds are strongly affecting the AMF, so their accuracy definitely matters.*

"Examples of cloud and NO$_2$ retrieval are shown in Figure A1. The O$_2$-O$_2$ and FRESCO cloud fraction retrievals show very good agreement. However, cloud pressure retrievals show large differences, especially for high cloud cases. It should be noted that the cloud pressure retrievals based on O$_2$-O$_2$ or O$_2$ absorption must be interpreted effective values. The accuracy of the cloud retrieval does not always link to the accuracy of the cloud correction in the NO$_2$ retrieval. For instance, the O$_2$-O$_2$ cloud pressure substantially differ from true values for the high cloud cases, whereas FRESCO cloud pressures are usually compared to the middle of the cloud layer. On the other hand, NO$_2$ AMF using an O$_2$-O$_2$ correction are often closer to the true AMF than those using FRESCO correction."

Rephrased by:

"Examples of cloud and NO$_2$ retrieval are shown in Figure A1. The O$_2$-O$_2$ and FRESCO cloud fraction retrievals show very good agreement. However, cloud pressure retrievals show large differences, especially for high cloud cases. It should be noted that cloud pressure retrievals based on O$_2$-O$_2$ or O$_2$ absorption must be interpreted as effective values. Furthermore, a more accurate cloud retrieval does not always correspond to a better cloud correction in the NO$_2$ retrieval. For instance, the O$_2$-O$_2$ cloud pressure substantially differs from true values for the high cloud cases, whereas FRESCO cloud pressures are usually compared to the middle of the cloud layer. On the other hand, NO$_2$ AMFs using an O$_2$-O$_2$ correction are often closer to the true AMF than those using a FRESCO correction. These results also show different impact on the retrieval between the polluted and clean cases. It implies that the accuracy of the cloud correction relies not only on the accuracy of the cloud retrieval, but also on other factors, such as the NO$_2$ profile."

*I also wonder if the authors have understood this fundamental problem, as in their reply, they refer to additional figures that show solely 1d cloud retrieval results. I don't understand how this 1d analysis makes the authors "believe that the analysis of 3D effects on cloud products is not a relevant topic in this paper".*

*So I still see a problem in the study design: 3d effects are solely analysed with respect to trace gas AMFs, but not on the cloud retrievals.*

*Thus the authors should clearly state this inconsistency in their study in abstract and introduction, and should discuss how far this inconsistency could be resolved in future studies in the discussion/conclusion.*

The aim of this study is to assess the performances/limitation of current trace gas retrievals. Therefore, cloud correction approaches used in the retrieval are based on simplified cloud correction schemes for all cloud effects, including 3D effect.

1D cloud/$NO_2$ retrievals show that the accuracy of the cloud corrections may depend on many factors, and it's therefore difficult to assess the accuracy of the cloud correction only based on an analysis of the accuracy of the cloud retrievals.

In addition, some descriptions of cloud product used for $NO_2$ retrieval and cloud retrievals in cloud shadow/in-scattering regions are added:

Introduction (Line 86): "The present paper focuses on impact of 3D effects on the classic tropospheric trace gas retrievals, including identification and investigation of the significant retrieval biases due to the 3D clouds, and exploration of mitigation strategies for these cases."

Methodologies (Line 126): "Notice that, all cloud effects, including the 3D effect, are treated based on such simplified cloud correction schemes, however, these approaches may not capture all cloud effects, which leads to uncertainty in the $NO_2$ retrieval."

NO2 retrieval in the vicinity of a box-cloud (Line 268): "For these pixels, the retrieved CFr is greater than 0 due to the enhanced reflectance, and the $O_2$-$O_2$ value is slightly higher than that of FRESCO. Cloud pressure retrieval is usually a bit lower than surface pressure, but higher than neighboring cloud pressure, and the FRESCO cloud pressure is relatively higher (not shown)."

Conclusions and Outlook (Line 566): "The cloud products used in the $NO_2$ retrieval treat the cloud shadow pixels as cloud-free, resulting in large positive biases (up to more than 100%) in the $NO_2$ AMF calculation."

Future work includes (Line 585): "Another mitigation method is to develop more sophisticated cloud retrievals, which account for the 3D effects, are feasible to apply to satellite observation, and can easily adapt to trace gas retrieval algorithms"

**Bibliography**

Boersma, K. F., Eskes, H. J., & Brinksma, E. J. (2004). Error analysis for tropospheric NO2 retrieval from space. *Journal of Geophysical Research: Atmospheres*, *109*(4). https://doi.org/10.1029/2003jd003962

Boersma, K. F., Eskes, H. J., Dirksen, R. J., Van Der A, R. J., Veefkind, J. P., Stammes, P., Huijnen, V., Kleipool, Q. L., Sneep, M., Claas, J., Leitão, J., Richter, A., Zhou, Y., & Brunner, D. (2011). An improved tropospheric NO2 column retrieval algorithm for the Ozone Monitoring Instrument. *Atmospheric Measurement Techniques*, *4*(9). https://doi.org/10.5194/amt-4-1905-2011

---

## Author Response (AR3)

We gratefully thank the reviewer for the careful reading of our revised manuscript and for the constructive comments. Below the reviewer's comments are reproduced in italic bold font. Our responses to these comments are given in roman font and pieces of text added to the manuscript are displayed in blue font.

**Referee :**

*The abstract, introduction and conclusions need to be updated to properly address the comments from Referee#2: "So I still see a problem in the study design: 3d effects are solely analysed with respect to trace gas AMFs, but not on the cloud retrievals. Thus the authors should clearly state this inconsistency in their study in abstract and introduction, and should discuss how far this inconsistency could be resolved in future studies in the discussion/conclusion."*

*The abstract was not updated at all, the changes in the introduction and conclusions (Line 566) do not mention the inconsistency.*

This has been added in the following parts:

Abstract (line 7):

"Although clouds have significant effects on trace gas retrievals, the current cloud correction schemes are based on a simple cloud model, and the retrieved cloud parameters must be interpreted as effective values. Consequently, it is difficult to assess the accuracy of the cloud correction only based on analysis of the accuracy of the cloud retrievals, and this study focuses solely on the impact of the 3D cloud structures on the trace gas retrievals."

Introduction (line 89):

"The 3D effects affect the cloud retrievals first and then the trace gas retrievals, and in this study, the main focus is on the influence of 3D clouds on the trace gas retrievals."

Conclusion (line 579):

"Since the cloud correction schemes are based on a simple cloud model, the accuracy of the $NO_2$ retrieval depends on not only the cloud retrieval, but also on other factors, such as the $NO_2$ profile. The analysis in the study focused mainly on the error of the NO2 retrieval due to the 3D cloud effects."